# A Provable Approach for End-to-End Safe Reinforcement Learning

**Akifumi Wachi**[*]          **Kohei Miyaguchi**[*]

**Takumi Tanabe**[*]          **Rei Sato**[*]          **Youhei Akimoto**[†‡]

[*]LY Corporation  [†]University of Tsukuba  [‡]RIKEN AIP
{akifumi.wachi, kmiyaguc, takumi.tanabe, sato.rei}@lycorp.co.jp
akimoto@cs.tsukuba.ac.jp

## Abstract

A longstanding goal in safe reinforcement learning (RL) is a method to ensure the safety of a policy throughout the entire process, from learning to operation. However, existing safe RL paradigms inherently struggle to achieve this objective. We propose a method, called Provably Lifetime Safe RL (PLS), that integrates offline safe RL with safe policy deployment to address this challenge. Our proposed method learns a policy offline using return-conditioned supervised learning and then deploys the resulting policy while cautiously optimizing a limited set of parameters, known as target returns, using Gaussian processes (GPs). Theoretically, we justify the use of GPs by analyzing the mathematical relationship between target and actual returns. We then prove that PLS finds near-optimal target returns while guaranteeing safety with high probability. Empirically, we demonstrate that PLS outperforms baselines both in safety and reward performance, thereby achieving the longstanding goal to obtain high rewards while ensuring the safety of a policy throughout the lifetime from learning to operation.

## 1 Introduction

Reinforcement learning (RL) has exhibited remarkable capabilities in a wide range of real problems, including robotics [32], data center cooling [34], finance [23], and healthcare [60]. RL has attracted significant attention through its successful deployment in language models [21, 38] or diffusion models [7]. As RL becomes a core component of advanced AI systems that affect our daily lives, ensuring the safety of these systems has emerged as a critical concern. Hence, while harnessing the immense potential of RL, we must simultaneously address and mitigate safety concerns [4].

Safe RL [18, 20] is a fundamental and powerful paradigm for incorporating explicit safety considerations into RL. Given its wide range of promising real-world applications, safe RL naturally spans a broad scope and involves several critical considerations in its formulation. For example, design choices must be made regarding the desired level of safety (e.g., safety guarantees are required in expectation or with high probability), the phase in which safety constraints are enforced (e.g., post-convergence or even during training), and other related aspects [27, 55].

A longstanding goal in safe RL is to develop a methodology with a safety guarantee throughout the entire process, from learning to operation. However, existing safe RL paradigms inherently struggle to achieve this goal. In online safe RL, where an agent learns its policy while interacting with the environment, ensuring safety is especially challenging during the initial phases of policy learning. While safe exploration [51], sim-to-real safe RL [24], or end-to-end safe RL [11] have been actively

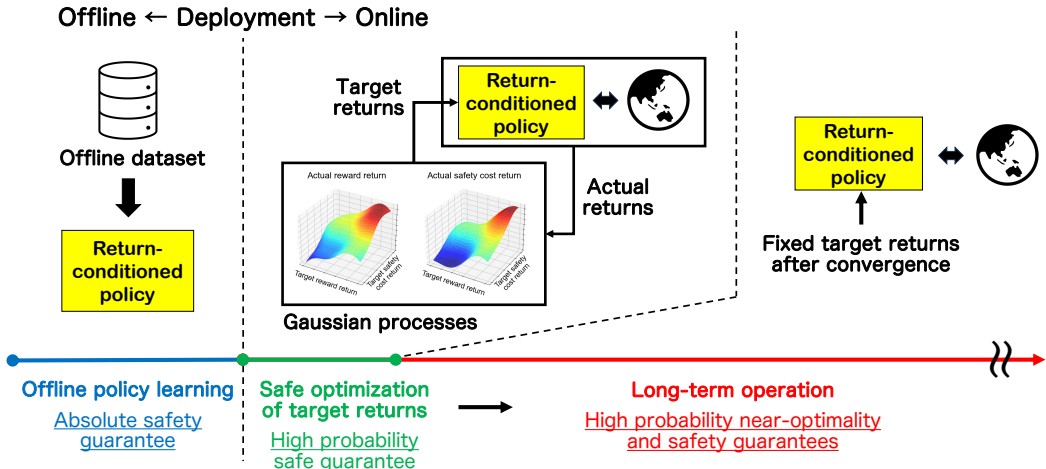

Figure 1: A conceptual illustration of PLS. After learning a return-conditioned policy using offline safe RL, PLS optimizes target returns through safe online policy evaluation via Gaussian processes. A key advantage of PLS is that safety is guaranteed at least with high probability in the entire process.

studied, they typically rely on strong assumptions, such as (partially) known state transitions. Also, in offline safe RL, where a policy is learned from a pre-collected dataset, it remains difficult to deploy a safe policy in a real environment due to distribution mismatch issues between the offline data and the actual environment, even though training can proceed without incurring any immediate safety risks.

**Our contributions.**   We propose Provably Lifetime Safe RL (PLS), an algorithm designed to address the longstanding goal in safe RL. PLS integrates *offline* policy learning with *online* policy evaluation and adaptation with high probability safety guarantee, as illustrated in Figure 1. Specifically, PLS begins by training a policy using an offline safe RL algorithm based on return-conditioned supervised learning (RCSL). Given this resulting return-conditioned policy, PLS then seeks to optimize a set of target returns by maximizing the reward return subject to a safety constraint during actual environmental interaction. Through rigorous analysis, we demonstrate that leveraging Gaussian processes (GPs) for this optimization is theoretically sound, which enables PLS to optimize target returns in a Bayesian optimization framework. We further prove that, with high probability, the resulting target returns are near-optimal while guaranteeing safety. Finally, empirical results demonstrate that 1) PLS outperforms baselines in both safety and task performance, and 2) PLS learns a policy that achieves high rewards while ensuring safety throughout the entire process from learning to operation.

## 2   Related Work

Safe RL [18] is a promising approach to bridge the gap between RL and critical decision-making problems related to safety. A constrained Markov decision process (CMDP, [3]) is a popular model for formulating a safe RL problem. In this problem, an agent must maximize the expected cumulative reward while guaranteeing that the expected cumulative safety cost is less than a fixed threshold.

**Online safe RL.** Although safe RL in CMDP settings has been substantially investigated, most of the existing literature has considered "online" settings, where the agent learns while interacting with the environment [55]. Prominent algorithms fall into this category, as represented by constrained policy optimization (CPO, [1]), Lagrangian-based actor-critic [6, 8], and primal-dual policy optimization [39, 58]. In online safe RL, satisfaction of safety constraints is not usually guaranteed during learning, and many unsafe actions may be executed before converging. To mitigate this issue, researchers have investigated safe exploration [5, 51, 53], formal methods [2, 17], or end-to-end safe RL [11, 25]. These techniques, however, typically rely on strong assumptions (e.g., known state transitions), and excessively conservative policies tend to result in unsatisfactory performance or inapplicability to complex systems. Therefore, simultaneously achieving both reward performance and guaranteed safety within the online safe RL paradigm is inherently difficult.

**Offline safe RL.** Offline reinforcement learning (RL) [33, 40] trains an agent exclusively on a fixed dataset of previously collected experiences. Since the agent does not interact with the environment during training, no potentially unsafe actions are executed during learning. Extending this setup to incorporate explicit safety requirements has led to the area of offline safe RL [30, 31, 37, 42, 56]. In this context, the objective is to maximize expected cumulative reward while satisfying pre-specified safety constraints, all from a static dataset. Because the policy is never deployed during training, offline safe RL is especially appealing for safety-critical domains. Le et al. [30] pioneered this direction with an algorithm that optimizes return under safety constraints using only offline data. Liu et al. [37] proposed a constrained decision transformer (CDT) that solves safe RL problems by sequence modeling by extending decision transformer [10] architectures from unconstrained to constrained RL settings. Despite such progress, offline safe RL still suffers from a central difficulty: learned policies often become either unsafe or overly conservative, largely due to the intrinsic challenges of off-policy evaluation (OPE) in stateful environments [15].

**Versatile safe RL.** Our PLS is also related to *versatile* safe RL, where an agent needs to incorporate a set of thresholds rather than a single predefined value. For example, in online safe RL settings, Yao et al. [59] proposes a framework called constraint-conditioned policy optimization (CCPO) that consists of versatile value estimation for approximating value functions under unseen threshold conditions and conditioned variational inference for encoding arbitrary constraint thresholds during policy optimization. Also, Lin et al. [35] proposes an algorithm to address offline safe RL problems with real-time budget constraints. Finally, Guo et al. [22] proposes an algorithm called constraint-conditioned actor-critic (CCAC) that models the relations between state-action distributions and safety constraints and then handles out-of-distribution data and adapts to varying constraint thresholds.

# 3 Problem Statement

We consider a sequential decision-making problem in a finite-horizon constrained Markov decision process (CMDP, [3]) defined as a tuple

$$\mathcal{M} \coloneqq \langle \mathcal{S}, \mathcal{A}, P, H, s_1, r, g \rangle, \tag{1}$$

where $\mathcal{S}$ is a state space, $\mathcal{A}$ is an action space, and $P : \mathcal{S} \times \mathcal{A} \to \Delta(\mathcal{S})$ is the state transition probability, where $\Delta(X)$ denotes the probability simplex over the set $X$. For ease of notation, we define a transition kernel $P_T : \mathcal{S} \times \mathcal{A} \to \Delta(\mathbb{R}^2 \times \mathcal{S})$ associated with $\langle P, r, g \rangle$. Additionally, $H \in \mathbb{Z}_+$ is the fixed finite length of each episode, $s_1 \in \mathcal{S}$ is the initial state, $r : \mathcal{S} \times \mathcal{A} \to [0, 1]$ is the normalized reward function bounded in $[0, 1]$. While we assume that the initial state is fixed to $s_1$, our key ideas can be easily extended to the case of initial state distribution $\Delta(\mathcal{S})$. A key difference from a standard (unconstrained) MDP lies in the (bounded) safety cost function $g : \mathcal{S} \times \mathcal{A} \to [0, 1]$. For succinct notation, we use $s_t$ and $a_t$ to denote the state and action at time $t$, and then define $\xi_t \coloneqq (s_t, a_t, r_t, g_t)$ for all $t \in [H]$, where $r_t = r(s_t, a_t)$ and $g_t = g(s_t, a_t)$.

Episodes are defined as sequences of states, actions, rewards, and safety costs $\Xi \coloneqq \{\xi_t\}_{t=1}^H \in (\mathcal{S} \times \mathcal{A} \times \mathbb{R}^2)^H$, where $s_{t+1} \sim P(\cdot \mid s_t, a_t)$ for all $t \in [H]$. The $t$-th context $x_t$ of an episode refers to the partial history $x_t \coloneqq (\xi_1, \xi_2, \ldots, \xi_{t-1}, s_t)$ for $1 \leq t \leq H + 1$, where we let $s_{H+1} = \bot$ be a dummy state. Let $\mathcal{X}_t \coloneqq (\mathcal{S} \times \mathcal{A} \times \mathbb{R}^2)^{t-1} \times \mathcal{S}$ be the set of all $t$-th contexts and $\mathcal{X} \coloneqq \bigcup_{t=1}^H \mathcal{X}_t$ be the sets of all contexts at time steps $1 \leq t \leq H$.

We consider a context-dependent policy $\pi : \mathcal{X} \to \Delta(\mathcal{A})$ to map a context to an action distribution, subsequently identifying a joint probability distribution $\mathbb{P}^\pi$ on $\Xi$ such that $a_t \sim \pi(x_t)$ and $(r_t, g_t, s_{t+1}) \sim P_T(s_t, a_t)$ for all $t \in [H]$.[1] Given a trajectory $\tau = (\xi_1, \xi_2, \ldots, \xi_H)$, returns are given by $\widehat{R}(\tau) \coloneqq \sum_{t=1}^H r(s_t, a_t)$ for reward and $\widehat{G}(\tau) \coloneqq \sum_{t=1}^H g(s_t, a_t)$ for safety cost, respectively. We now define the following two metrics that are respectively called reward and safety cost returns, where the expectation is taken over trajectories $\tau$ induced by a policy $\pi$ and the transition kernel $P_T$:

$$J_r(\pi) = \mathbb{E}_{\tau \sim \pi, P_T}\left[\widehat{R}(\tau)\right] \quad \text{and} \quad J_g(\pi) = \mathbb{E}_{\tau \sim \pi, P_T}\left[\widehat{G}(\tau)\right]. \tag{2}$$

**Dataset.** We assume access to an offline dataset $\mathcal{D} \coloneqq \{\Xi^{(i)}\}_{i=1}^n$, where $n \in \mathbb{Z}_+$ is a positive integer. Let $\beta : \mathcal{X} \to \Delta(\mathcal{A})$ denote a behavior policy. The dataset $\mathcal{D}$ comprises $n$ independent episodes

---

[1]In this paper, we focus on *context-dependent* policies, a broader class than the state-dependent policies that dominate most prior RL work.

generated by $\beta$; that is, $\mathcal{D} \sim (\mathbb{P}^\beta)^n$. We also assume that, for any $x_t \in \mathcal{X}$, the behavior action distribution $\beta(x_t)$ is conditionally independent of past rewards $\{r_h\}_{h=1}^{t-1}$ and safety costs $\{g_h\}_{h=1}^{t-1}$ given past states and actions $x_t \setminus \{r_h, g_h\}_{h=1}^{t-1}$.

**Goal.** We solve a *versatile* safe RL problem in the CMDP, where the safety threshold $b$ is chosen within a set of candidate thresholds $\mathcal{B} := [0, H]$. Specifically, our goal is to optimize a single policy $\pi$ that maximizes $J_r(\pi)$ while ensuring that $J_g(\pi)$ is less than a threshold $b \in \mathcal{B}$:

$$\max_\pi \ J_r(\pi) \quad \text{subject to} \quad J_g(\pi) \leq b, \quad \forall b \in \mathcal{B}. \tag{3}$$

In contrast to the standard safe RL problems, we additionally address two fundamental and important challenges. First, our goal is to learn, deploy, and operate a policy for solving (3) while guaranteeing safety throughout the entire safe RL process from learning to operation, at least with high probability. Second, we aim to train a single policy that can adapt to diverse safety thresholds $b \in \mathcal{B}$.

# 4 Preliminaries

## 4.1 Return-Conditioned Supervised Learning

Return-conditioned supervised learning (RCSL) is a methodology to learn the return-conditional distribution of actions in each state and then define a policy by sampling from the action distribution with high returns. RCSL was first proposed in online RL settings [29, 43, 46] and was then extended to offline RL settings [10, 14]. In offline RL settings, RCSL aims at estimating the return-conditioned behavior (RCB) policy $\beta_R(a \mid x) := \mathbb{P}^\beta(a_t = a \mid x_t = x, \widehat{R} = R)$; that is, the action distribution conditioned on the return $\widehat{R} = R \in [0, H]$ and the context $x_t = x \in \mathcal{X}$. According to the Bayes' rule, the RCB policy $\beta_R : \mathcal{X} \to \Delta(\mathcal{A})$ is written as the importance-weighted behavior policy

$$d\beta_R(a \mid x) = f(R \mid x, a) / f(R \mid x) \cdot d\beta(a \mid x), \tag{4}$$

where $f(R \mid x) := \frac{d}{dR} \mathbb{P}^\beta(\widehat{R} \leq R \mid x_t = x)$ and $f(R \mid x, a) := \frac{d}{dR} \mathbb{P}^\beta(\widehat{R} \leq R \mid x_t = x, a_t = a)$ respectively denote the conditional probability density functions of the behavior return.[2]

## 4.2 Decision Transformer

Decision transformer (DT, [10]) is a representative instance of the RCSL. In DT, trajectories are modeled as sequences of states, actions, and returns (i.e., reward-to-go). DT policies are typically learned using the GPT architecture [41] with a causal self-attention mask; thus, action sequences are generated in an autoregressive manner. The pre-training of DT can be seen as a regularized maximum likelihood estimate (MLE) of the neural network parameters

$$\hat{\theta} = \underset{\theta \in \Theta}{\operatorname{argmin}} \left\{ -\frac{1}{nH} \sum_{i=1}^n \sum_{t=1}^H \ln p_\theta(a_t^{(i)} \mid x_t^{(i)}, \widehat{R}^{(i)}) + \Phi(\theta) \right\}, \tag{5}$$

where $\mathcal{P} := \{p_\theta(a \mid x, R)\}_{\theta \in \Theta}$ is a parametric model of conditional probability densities, and $\Phi(\theta) \geq 0$ is a penalty term representing inductive biases in parameter optimization. The output of DT is then given by $\pi_{\hat{\theta}, R}$, where $\pi_{\theta, R}$ denotes the policy associated with $p_\theta(\cdot \mid \cdot, R)$.

## 4.3 Constrained Decision Transformer

Constrained decision transformer (CDT, [37]) is a promising paradigm that extends the DT to constrained reinforcement learning by conditioning the policy on both reward and safety-cost returns. Specifically, CDT parameterizes a policy to take states, actions, reward returns, and safety cost returns as input tokens, and then generates the same length of predicted actions as output. Although practical implementations often truncate the input to a fixed context length, we simplify the analysis by assuming that the entire history $x_t$ is provided to the model.

---

[2]Strictly speaking, the right-hand side of (4) can be ill-defined for certain $x \in \mathcal{X}$ and $a \in \mathcal{A}$ if either $f(R \mid x)$ or $f(R \mid x, a)$ are ill-defined, or if $f(R \mid x) = 0$. For our analysis, however, it suffices to impose (4) on $\beta_R$ only when the right-hand side is well-defined.

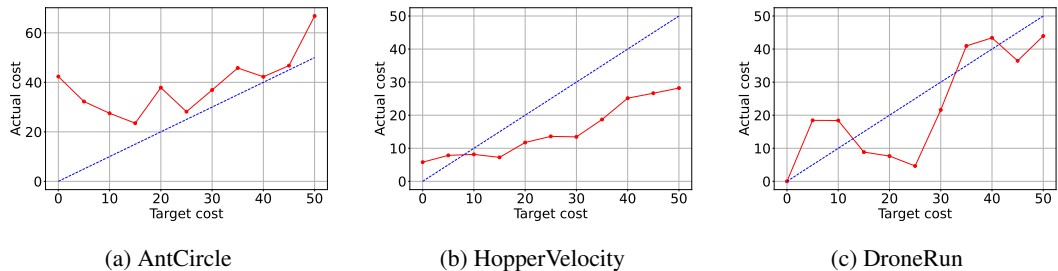

| (a) AntCircle | (b) HopperVelocity | (c) DroneRun |

Figure 2: Relations between target safety cost return $G$ and actual safety cost return $J_g(\pi)$ of pretrained CDT policies (red lines). Blue dotted lines represent $y = x$. Target reward returns are fixed with the reward returns of the best trajectories included in the offline dataset. Observe that CDT policies suffer from unsuccessful misalignment between actual returns and target returns: (a) constraint violation, (b) excessively conservative behavior, and (c) both.

In the inference phase, a user specifies a target reward return $R$ and target safety cost return $G$ at the beginning of the episode and iteratively update the target returns for the next time step by $R_{t+1} = R_t - r_t$ and $G_{t+1} = G_t - g_t$ with $R_1 = R$ and $G_1 = G$. Since the target returns play critical roles in the CDT framework, we explicitly add them in the notations of $\pi$ to emphasize the dependence on the pair of target returns $\boldsymbol{z} := (R, G)$; that is, let us denote $\pi_{\hat{\theta}, \boldsymbol{z}}(a \mid x)$ and define $\mathcal{Z}$ to be the set of all $\boldsymbol{z}$ that are feasible. Crucially, since CDT is a variant of RCSL that extends DT to constrained RL settings, the mathematical discussions are also true with CDT by replacing $R$ with $\boldsymbol{z}$, by defining $f(\boldsymbol{z} \mid x)$ in (4) or $p_\theta(\cdot \mid \cdot, R, G)$ in (5), for example.

**Safety issues of CDT policies.** Ideally, we desire to align actual returns with target returns; that is, $J_r(\pi_{\hat{\theta}, \boldsymbol{z}}) \approx R$ and $J_g(\pi_{\hat{\theta}, \boldsymbol{z}}) \approx G$ for $\boldsymbol{z} = (R, G)$. This is why the target reward return $R$ is typically set to be the maximum return included in the offline dataset, while the target safety cost return $G$ is set to be the safety threshold. Unfortunately, however, the actual returns are *not* necessarily aligned with the correct target returns. As evidence, Figure 2 shows the empirical relations between target returns and actual returns of CDT policies. Specifically, actual returns may differ from corresponding target returns, and their differences vary depending on the tasks or pre-trained CDT models.

## 5 Theoretical Relations Between Target and Actual Returns

In Figure 2, while we observe discrepancies between the target and actual returns, there seem to be some relations that can be captured using data. Our goal here is to theoretically understand when and how closely the CDT policy $\pi_{\hat{\theta}, \boldsymbol{z}}$ achieves the target returns, $\boldsymbol{z}$. Unfortunately, however, given the architecture and learning complexity of CDTs, it is almost impossible to conduct such theoretical analyses without any assumptions; hence, we first list several necessary assumptions.

**Assumption 1** (Near-deterministic transition). Let $\boldsymbol{q} := (r, g)$ denote a pair of reward and safety cost. Also, let $p_q(\boldsymbol{q}' \mid s, a) := \frac{\mathrm{d}}{\mathrm{d}\boldsymbol{q}'} P_T\{r \le r', g \le g' \mid s, a\}$ be the corresponding density function. There exist deterministic maps $\hat{\boldsymbol{q}}(\cdot, \cdot)$, $\hat{s}'(\cdot, \cdot)$, and small constants $\epsilon_q, \epsilon_s, \delta \ge 0$ such that $p_q(\boldsymbol{q} \mid s, a) \le \epsilon_q$ for all $\|\boldsymbol{q} - \hat{\boldsymbol{q}}(s, a)\|_\infty > \delta$ and $P\{s' \ne s'(s, a) \mid s, a\} \le \epsilon_s$ for all $s \in \mathcal{S}$ and $a \in \mathcal{A}$.

Assumption 1 is more general than that used in Brandfonbrener et al. [9] because 1) ours is for multiobjective settings and 2) we consider $\delta$-neighborhood rather than exact equality (i.e., $\delta = 0$). Especially, the second extension is beneficial since we can analyze theoretical properties of CDT policies optimized based on continuous reward and safety cost, whereas Brandfonbrener et al. [9] effectively limits the scope of application to the problems with discrete rewards. This is a significant extension because safe RL problems typically require the agent to deal with safety constraints with continuous safety cost functions and thresholds. Moreover, similar (near-)deterministic assumptions are common in notable safe RL literature [5, 50, 53].

We then make assumptions about the conditional probability density function of the behavior return; that is, $f$ defined in (4). With a slight extension from $R$ to $\boldsymbol{z}$, we assume the following three conditions on $f(\boldsymbol{z} \mid x)$, with $\boldsymbol{z}$ fixed to a value of interest.

**Assumption 2** (Initial coverage). $\eta_{\boldsymbol{z}} := f(\boldsymbol{z} \mid s_1) > 0$.

**Assumption 3** (Boundedness). $C_{\boldsymbol{z}} := \sup_{x \in \mathcal{X}} f(\boldsymbol{z} \mid x) < \infty.$

**Assumption 4** (Continuity). $c_{\boldsymbol{z}}(\delta) := \sup_{\boldsymbol{z}':\|\boldsymbol{z}'-\boldsymbol{z}\|_{\infty} \leq 2\delta, \, x \in \mathcal{X}} |f(\boldsymbol{z}' \mid x) - f(\boldsymbol{z} \mid x)| < \infty$ is small.

Finally, we assume the expressiveness and regularity of the regularized model $(\mathcal{P}, \Phi)$ in (5) to control the behavior of the MLE, $\hat{\theta}$. The following assumptions are fairly standard and borrowed from Van der Vaart [52]; therefore, for ease of understanding, we will make informal assumptions below. See Appendix D.3 for the formal presentations of these assumptions.

**Assumption 5** (Soft realizability, *informal*). There exists $\theta^* \in \Theta$ such that $\beta_R$ and $\pi_{\theta^*, R}$ are close to each other regarding the KL divergence and $\Phi(\theta^*)$ is small. See Assumption 14 for a formal version.

**Assumption 6** (Regularity, *informal*). $\mathcal{P}$ and $\Phi$ are 'regular' enough for $\hat{\theta}$ to be asymptotically normal. See Assumption 15 for a formal version.

Finally, we present a theorem that characterizes the relation between target and actual returns.

**Theorem 1** (Relation between target and actual returns). *For any policy $\pi$, let us define $\boldsymbol{J}(\pi) := (J_r(\pi), J_g(\pi))$. Also, let $\pi_{\hat{\theta}, \boldsymbol{z}}$ denote the policy obtained by the algorithm, which is characterized by a set of target returns $\boldsymbol{z} = (R, G)$. Recall that $n$ is the number of trajectories contained in the offline dataset. Then, under Assumptions 1 - 6, we have*

$$\left\| \boldsymbol{J}(\pi_{\hat{\theta}, \boldsymbol{z}}) - \boldsymbol{z} - \frac{H^2}{\sqrt{n}} \mathcal{F}(\boldsymbol{z}) \right\|_{\infty} \leq \varepsilon(\boldsymbol{z}) + o_P\left(\frac{1}{\sqrt{n}}\right), \tag{6}$$

*where $\varepsilon(\boldsymbol{z})$ is a small bias function and $\mathcal{F} : [0, H]^2 \to \mathbb{R}^2$ is a sample path of a Gaussian process $GP(0, \boldsymbol{k})$, whose precise definitions are given in Theorems 4 and 7, respectively. Here, $o_P(\cdot)$ is the probabilistic small-o notation, i.e., $b_n = o_P(a_n)$ implies $\lim_{n \to \infty} \mathbb{P}\{|b_n/a_n| > \epsilon\} = 0, \forall \epsilon > 0$.*

See Appendix E for its formal statement and complete proof. Intuitively, the difference between the target and actual returns is decomposed into an unbiased Gaussian process term $H^2 \mathcal{F}(\boldsymbol{z})/\sqrt{n}$, a small bias term $\varepsilon(\boldsymbol{z})$, and an asymptotically negligible term $o_P(1/\sqrt{n})$.

**Remark 1** (Smoothness). Examining the explicit form of the covariance function $\boldsymbol{k}(\cdot, \cdot)$ reveals that $\mathcal{F}(\cdot)$ is smooth (under suitable conditions). Specifically, the smoothness of $\mathcal{F}(\cdot)$ is known to be closely matches that of $\boldsymbol{k}$ (Corollary 1 in [13]). For more details, see Remark 9.

It is important to clarify the role of our assumptions. While Assumptions 1-6 are necessary for the rigorous analysis in Theorem 1, we designed the PLS framework itself to be a more general meta-algorithm. Our experiments show that PLS remains robust and safe even when these assumptions, such as near-deterministic transitions, do not strictly hold. This suggests the core framework is applicable well beyond the conditions required for the theoretical guarantees.

## 6 Provably Lifetime Safe Reinforcement Learning

We finally present Provably Lifetime Safe Reinforcement Learning (PLS), a simple yet powerful approach that advances safe RL toward the longstanding goal of end-to-end safety.

As illustrated in Figure 1, PLS begins with offline policy learning from a pre-collected dataset. Since RL agents are most prone to violating safety constraints during the early phases of learning, this offline learning step is particularly beneficial for ensuring lifetime safety. Also, a key idea behind PLS is the use of a constrained RCSL (e.g., CDT) for this offline policy learning step. This approach yields a return-conditioned policy that enables control over both reward and safety performance through a few significant parameters. In the case of a single safety constraint, all we have to do is optimize a two-dimensional target return vector. Therefore, this method offers several advantages, including computational efficiency and enhanced controllability of policy behavior.

Hereinafter, we suppose there is a pre-trained policy obtained by constrained RCSL. For simplicity, we denote such a return-conditioned policy as $\pi_{\boldsymbol{z}}$ characterized by target reward and safety cost returns $\boldsymbol{z} = (R, G)$ while omitting the neural network parameters $\hat{\theta}$.

### 6.1 Characterizing Reward and Safety Cost Returns via Gaussian Processes

Guided by Theorem 1, we employ GPs to model the mapping from a target return vector $\boldsymbol{z} = (R, G)$ to the actual returns $\boldsymbol{J}(\pi_{\boldsymbol{z}}) := (J_r(\pi_{\boldsymbol{z}}), J_g(\pi_{\boldsymbol{z}}))$. We formulate this as a supervised learning problem

with the dataset $\{(\boldsymbol{z}_j, \boldsymbol{J}(\pi_{\boldsymbol{z}_j}))\}_{j=1}^N$, where $\boldsymbol{z}_1, \boldsymbol{z}_2, \dots, \boldsymbol{z}_N \in \mathcal{Z}$ is a sequence of target returns. For tractability, we discretize the search space, yielding a finite candidate set $\mathcal{Z}$ with cardinality $|\mathcal{Z}|$. While collecting such data, we sequentially choose the next target returns $\boldsymbol{z} \in \mathcal{Z}$ that maximize the actual reward return $J_r(\pi_{\boldsymbol{z}})$ subject to the safety constraint (i.e., $J_g(\pi_{\boldsymbol{z}}) \leq b$). The measured returns are assumed to be perturbed by i.i.d. Gaussian noise for sampled inputs $Z_N := [\boldsymbol{z}_1, \dots, \boldsymbol{z}_N]^\top \subseteq \mathcal{Z}$. Thus, for $\diamond \in \{r, g\}$ (the symbol $\diamond$ is used as a wildcard), we model the noise-perturbed observations by $y_{\diamond,j} = J_\diamond(\pi_{\boldsymbol{z}_j}) + w_{\diamond,j}$ with $w_{\diamond,j} \sim \mathcal{N}(0, \nu_\diamond^2)$, for all $j \in [N]$.

A GP is a stochastic process that is fully specified by a mean function and a kernel. We model the reward and safety cost returns with separate GPs:

$$J_r(\pi_{\boldsymbol{z}}) \sim \mathsf{GP}(\mu_r(\boldsymbol{z}), k_r(\boldsymbol{z}, \tilde{\boldsymbol{z}})) \quad \text{and} \quad J_g(\pi_{\boldsymbol{z}}) \sim \mathsf{GP}(\mu_g(\boldsymbol{z}), k_g(\boldsymbol{z}, \tilde{\boldsymbol{z}})), \tag{7}$$

where $\mu_\diamond(\boldsymbol{z})$ is a mean function and $k_\diamond(\boldsymbol{z}, \tilde{\boldsymbol{z}})$ is a covariance function for $\diamond \in \{r, g\}$. In principle, $J_r(\pi_{\boldsymbol{z}})$ and $J_g(\pi_{\boldsymbol{z}})$ may be correlated (i.e., off-diagonal elements in $\boldsymbol{k}$ is non-zero in Theorem 1), but we ignore these cross-correlations and learn each GP independently for simplicity.

Then, given the previous inputs $Z_N = [\boldsymbol{z}_1, \dots, \boldsymbol{z}_N]^\top$ and observations $\boldsymbol{y}_{\diamond,N} := \{y_{\diamond,1}, \dots, y_{\diamond,N}\}$, we can analytically compute a GP posterior characterized by the the mean $\mu_{\diamond,N}(\boldsymbol{z}) = \boldsymbol{k}_{\diamond,N}(\boldsymbol{z})^\top (\boldsymbol{K}_{\diamond,N} + \nu_\diamond^2 \mathbb{I}_N)^{-1} \boldsymbol{y}_{\diamond,N}$ and variance $\sigma_{\diamond,N}^2(\boldsymbol{z}) = k_\diamond(\boldsymbol{z}, \boldsymbol{z}) - \boldsymbol{k}_{\diamond,N}(\boldsymbol{z})^\top (\boldsymbol{K}_{\diamond,N} + \nu_\diamond^2 \mathbb{I}_N)^{-1} \boldsymbol{k}_{\diamond,N}(\boldsymbol{z})$, where $\boldsymbol{k}_{\diamond,N}(\boldsymbol{z}) = [k_\diamond(\boldsymbol{z}_1, \boldsymbol{z}), \dots, k_\diamond(\boldsymbol{z}_N, \boldsymbol{z})]^\top$ and $\boldsymbol{K}_{\diamond,N}$ is the positive definite kernel matrix $[k_\diamond(\boldsymbol{z}, \tilde{\boldsymbol{z}})]_{\boldsymbol{z}, \tilde{\boldsymbol{z}} \in Z_N}$, and $\mathbb{I}_N \in \mathbb{R}^{N \times N}$ is the identify matrix. Finally, we assume that $J_g(\pi_{\boldsymbol{z}})$ is $L$-Lipschitz continuous with respect to some distance metric $d(\cdot, \cdot)$ in $\mathcal{Z}$. This assumption is rather mild and is automatically satisfied by many commonly-used kernels [45, 48].

## 6.2 Safe Exploration and Optimization of Target Returns

Our current goal is to find the optimal pair of target returns $\boldsymbol{z} = (R, G)$ that maximizes $J_r(\pi_{\boldsymbol{z}})$ while guaranteeing the satisfaction of the safety constraint (i.e., $J_g(\pi_{\boldsymbol{z}}) \leq b$) according to GP-based inferences. For this purpose, we optimistically sample the next target returns $\boldsymbol{z}$ while pessimistically ensuring the satisfaction of the safety constraint, as conducted in Sui et al. [49].

A key advantage of using GPs is that we can estimate the uncertainty of the actual returns $J_r$ and $J_g$. To guarantee, high probability, both constraint satisfaction and reward maximization, for each function $\diamond \in \{r, g\}$, we construct a confidence interval defined as $\Omega_{\diamond,N}(\boldsymbol{z}) := [\mu_{\diamond,N-1}(\boldsymbol{z}) \pm \alpha_{\diamond,N} \cdot \sigma_{\diamond,N-1}(\boldsymbol{z})]$, where $\alpha_{\diamond,N} \in \mathbb{R}_+$ is a positive scalar that balances exploration and exploitation. These coefficients $\alpha_r$ and $\alpha_g$ are crucial in the performance of PLS, and principled choices for these coefficients have been extensively studied in the Bayesian optimization literature (e.g., [12, 45]). Thus, following Srinivas et al. [45], we define

$$\alpha_{r,j} = \alpha_{g,j} = \sqrt{2 \log(|\mathcal{Z}| \, j^2 \Pi^2 / (6\Delta))}, \tag{8}$$

where $\Delta \in [0, 1]$ is the allowed failure probability, and $\Pi$ in (8) is the circle ratio, not a policy.

To expand the set of feasible target returns $\boldsymbol{z}$ while satisfying the safety constraint, we use alternative confidence intervals $\Lambda_N(\boldsymbol{z}) := \Lambda_{N-1}(\boldsymbol{z}) \cap \Omega_{g,N}(\boldsymbol{z})$ with $\Lambda_0(\boldsymbol{z}) = [0, b]$ so that $\Lambda_N$ are sequentially contained in $\Lambda_{N-1}$ for all $N$. We thus define an upper bound $u_N(\boldsymbol{z}) := \max \Lambda_N(\boldsymbol{z})$ and a lower bound of $\ell_N(\boldsymbol{z}) := \min \Lambda_N(\boldsymbol{z})$, respectively. Note that $u_N$ is monotonically non-increasing and $\ell_N$ is monotonically non-decreasing, with respect to $N$.

**Safe exploration.** Using the GP upper confidence bound, we construct the set of *safe* target returns by $\mathcal{Y}_N = \bigcup_{\boldsymbol{z} \in \mathcal{Y}_{N-1}} \{\boldsymbol{z}' \in \mathcal{Z} \mid u_N(\boldsymbol{z}) + L \cdot d(\boldsymbol{z}, \boldsymbol{z}') \leq b\}$. At each iteration, PLS computes a set of $\boldsymbol{z}$ that are likely to increase the number of candidates for safe target returns. The agent thus picks $\boldsymbol{z}$ with the highest uncertainty while satisfying the safety constraint with high probability; that is,

$$\boldsymbol{z}_N = \operatorname*{argmax}_{\boldsymbol{z} \in E_N} (u_N(\boldsymbol{z}) - \ell_N(\boldsymbol{z})) \quad \text{with} \quad E_N = \{\boldsymbol{z} \in \mathcal{Y}_N : e_N(\boldsymbol{z}) > 0\}, \tag{9}$$

where $e_N(\boldsymbol{z}) := |\{\boldsymbol{z}' \in \mathcal{Z} \setminus \mathcal{Y}_N \mid \ell_N(\boldsymbol{z}) - L \cdot d(\boldsymbol{z}, \boldsymbol{z}') \leq b\}|$. Intuitively, $e_N(\cdot)$ optimistically quantifies the potential enlargement of the current safe set after obtaining a new sample $\boldsymbol{z}$.

**Reward maximization.** Safe exploration is terminated under the condition $\max_{\boldsymbol{z} \in E_N} (u_N(\boldsymbol{z}) - \ell_N(\boldsymbol{z})) \leq \zeta$, where $\zeta \in \mathbb{R}_+$ is a tolerance parameter. After fully exploring the set of safe target

returns, we turn to maximizing $J_r(\cdot)$ under the safety constraint. Concretely, we choose the next target returns optimistically within the pessimistically constructed set of safe target returns by

$$z_N = \underset{z \in \mathcal{Y}_N}{\arg\max} \big( \mu_{r,N}(z) + \alpha_{r,N} \cdot \sigma_{r,N}(z) \big). \tag{10}$$

### 6.3 Theoretical Guarantees on Safety and Near-optimality

We provide theoretical results on the overall properties of PLS. We will make an assumption and then present two theorems on safety and near-optimality. The assumption below is fairly mild in practice, because we can easily ensure that the return-conditioned policy meets the safety constraint by conservatively choosing small target returns, $R$ and $G$. See Appendix J for the full proofs.

**Assumption 7** (Initial safe set). *There exists a singleton seed set $\mathcal{Z}_0$ that is known to satisfy the safety constraint; that is, for all $z \in \mathcal{Z}_0$, $J_g(\pi_z) \leq b$ holds.*

**Theorem 2** (Safety guarantee). *At every iteration $j$, suppose that $\alpha_{g,j}$ is set as in (8) and the target returns $z_j$ are chosen within $\mathcal{Y}_j$. Then, $J_g(\pi_{z_j}) \leq b$ holds — i.e., the safety constraint is satisfied — for all $j \geq 0$, with a probability of at least $1 - \Delta$ .*

Intuitively, because PLS samples the next target returns $z$ so that the GP upper bound $u(z)$ is smaller than the threshold $b$, the true value $J_g(\pi_z)$ is guaranteed to be smaller than $b$ with high probability under proper assumptions. Moreover, since PLS learns the return-conditioned policy *offline*, Theorem 2 leads to an end-to-end safety guarantee, ensuring that the constraint is satisfied from learning to operation, with at least a high probability.

**Theorem 3** (Near-optimality). *Set $\alpha_{r,j}$ as in (8) for all $j \geq 0$. Let $z^\star$ denote the optimal feasible target returns. For any $\mathcal{E} \geq 0$, define $N_\sharp$ as the smallest positive integer $N$ satisfying*

$$4\sqrt{C_\nu \xi_{r,N} N^{-1} \log\big( |\mathcal{Z}| \Pi^2 N^2 / (6\Delta) \big)} \leq \mathcal{E}, \tag{11}$$

*where $C_\nu := 1 / \log(1 + \nu_r^{-2})$. Then, PLS finds a near-optimal $z$ such that:*

$$J_r(\pi_z) \geq J_r(\pi_{z^\star}) - \mathcal{E} \tag{12}$$

*with a probability at least $1 - \Delta$, after collecting $N_\sharp$ GP observations for reward maximization.*

Theorem 3 characterizes the online sample complexity of PLS. Following the analysis of Sui et al. [48], we can show that the safe exploration phase expands the estimated safe set until it contains the optimal target return vector $z^\star$ after at most $N_\dagger \in \mathbb{Z}_+$ GP iterations. Consequently, Theorem 3 thus implies that PLS will find a near-optimal target return vector $z$ using at most $\varpi(N_\dagger + N_\sharp)$ trajectories, where $\varpi \in \mathbb{Z}_+$ is the number of trajectories used for sample approximations of $J_r$ and $J_g$ for each GP update. Because PLS optimizes only the two-dimensional target return vector (i.e., $R$ and $G$), it requires far fewer online interactions than conventional online safe RL algorithms, which is an essential advantage in safety-critical settings where every interaction is costly or risky.

## 7 Experiments

We conduct empirical experiments for evaluating our PLS in multiple continuous robot locomotion tasks designed for safe RL. We adopt Bullet-Safety-Gym [19] and Safety-Gymnasium [26] benchmarks and implement our PLS and baseline algorithms using OSRL and DSRL libraries [36]. Experimental details are deferred to Appendix K.

**Metrics.** Our evaluation metrics are reward return and safety cost return, respectively normalized by $\widehat{R}_{\text{normalized}}(\pi) := \frac{\widehat{R}(\pi) - R_{\min,b}^\dagger}{R_{\max,b}^\dagger - R_{\min,b}^\dagger}$ and $\widehat{G}_{\text{normalized}}(\pi) := \frac{\widehat{G}(\pi)}{b}$. Recall that $\widehat{R}(\pi)$ and $\widehat{G}(\pi)$ are defined as the evaluated cumulative reward and safety cost that are obtained by a policy $\pi$. In the above definitions, $R_{\max,b}^\dagger$ and $R_{\min,b}^\dagger$ are the maximum and minimum cumulative rewards of the trajectories in the offline dataset $\mathcal{D}$. Note that we call a policy safe if $\widehat{G}_{\text{normalized}}(\pi) \leq 1$.

**Baselines.** We compare PLS against the following six baseline algorithms: BCQ-Lag, BEAR-Lag, CPQ, COptiDICE, CDT, and CCAC. BCQ-Lag and BEAR-Lag are both Lagrangian-based methods that apply PID-Lagrangian [47] to BCQ [16] and BEAR [28], respectively. CPQ [57] is an offline safe

Table 1: Experimental result with the safety cost threshold $b = 20$. The mean and standard deviation over 5 runs for each algorithm are shown. Reward and cost are normalized. **Bold**: Safe agents whose normalized cost is smaller than 1. Red: Unsafe agents. Blue: Safe agent with the highest reward.

| Task | Metric | BCQ-Lag | BEAR-Lag | CPQ | COptiDICE | CDT | CCAC | PLS |
|---|---|---|---|---|---|---|---|---|
| Ant-Run | Reward ↑ | 0.79 ± 0.05 | 0.07 ± 0.02 | 0.01 ± 0.01 | 0.63 ± 0.01 | 0.72 ± 0.05 | 0.02 ± 0.00 | 0.78 ± 0.06 |
| | Safety cost ↓ | 5.52 ± 0.67 | 0.12 ± 0.13 | 0.00 ± 0.00 | 0.79 ± 0.42 | 0.90 ± 0.12 | 0.00 ± 0.00 | 0.77 ± 0.10 |
| Ant-Circle | Reward ↑ | 0.59 ± 0.18 | 0.58 ± 0.24 | 0.00 ± 0.00 | 0.16 ± 0.13 | 0.47 ± 0.00 | 0.62 ± 0.13 | 0.41 ± 0.01 |
| | Safety cost ↓ | 2.28 ± 1.50 | 3.37 ± 1.71 | 0.00 ± 0.00 | 2.98 ± 3.55 | 2.23 ± 0.00 | 1.24 ± 0.55 | 0.77 ± 0.05 |
| Car-Circle | Reward ↑ | 0.65 ± 0.19 | 0.76 ± 0.12 | 0.70 ± 0.03 | 0.48 ± 0.04 | 0.73 ± 0.01 | 0.72 ± 0.03 | 0.72 ± 0.01 |
| | Safety cost ↓ | 2.17 ± 1.10 | 2.74 ± 0.89 | 0.01 ± 0.07 | 1.85 ± 1.48 | 0.98 ± 0.12 | 0.87 ± 0.29 | 0.88 ± 0.09 |
| Drone-Run | Reward ↑ | 0.65 ± 0.11 | -0.03 ± 0.02 | 0.19 ± 0.01 | 0.69 ± 0.03 | 0.57 ± 0.00 | 0.82 ± 0.05 | 0.59 ± 0.00 |
| | Safety cost ↓ | 3.91 ± 2.02 | 0.00 ± 0.00 | 0.00 ± 0.00 | 3.48 ± 0.19 | 0.34 ± 0.29 | 7.62 ± 0.37 | 0.50 ± 0.44 |
| Drone-Circle | Reward ↑ | 0.69 ± 0.05 | 0.82 ± 0.06 | -0.26 ± 0.01 | 0.22 ± 0.10 | 0.60 ± 0.00 | 0.37 ± 0.14 | 0.59 ± 0.00 |
| | Safety cost ↓ | 1.92 ± 0.64 | 3.58 ± 0.74 | 0.14 ± 0.39 | 0.68 ± 0.46 | 1.12 ± 0.06 | 0.74 ± 0.24 | 0.90 ± 0.08 |
| Ant-Velocity | Reward ↑ | 1.00 ± 0.01 | -1.01 ± 0.00 | -1.01 ± 0.00 | 1.00 ± 0.01 | 0.97 ± 0.00 | 0.68 ± 0.34 | 0.98 ± 0.00 |
| | Safety cost ↓ | 3.22 ± 0.60 | 0.00 ± 0.00 | 0.00 ± 0.00 | 6.60 ± 1.07 | 0.36 ± 0.22 | 0.60 ± 0.21 | 0.82 ± 0.19 |
| Walker2d -Velocity | Reward ↑ | 0.78 ± 0.00 | 0.89 ± 0.04 | -0.02 ± 0.03 | 0.13 ± 0.01 | 0.80 ± 0.00 | 0.81 ± 0.07 | 0.79 ± 0.00 |
| | Safety cost ↓ | 0.44 ± 0.32 | 7.60 ± 2.89 | 0.00 ± 0.00 | 1.75 ± 0.31 | 0.01 ± 0.04 | 6.37 ± 0.95 | 0.00 ± 0.00 |
| HalfCheetah -Velocity | Reward ↑ | 1.03 ± 0.03 | 0.98 ± 0.03 | 0.22 ± 0.33 | 0.63 ± 0.01 | 0.96 ± 0.03 | 0.84 ± 0.01 | 0.99 ± 0.00 |
| | Safety cost ↓ | 27.00 ± 8.76 | 12.35 ± 8.63 | 0.28 ± 0.23 | 0.00 ± 0.00 | 0.03 ± 0.13 | 1.36 ± 0.19 | 0.15 ± 0.19 |
| Hopper -Velocity | Reward ↑ | 0.85 ± 0.22 | 0.36 ± 0.11 | 0.20 ± 0.00 | 0.14 ± 0.10 | 0.68 ± 0.06 | 0.17 ± 0.09 | 0.83 ± 0.01 |
| | Safety cost ↓ | 8.48 ± 2.75 | 10.39 ± 3.79 | 3.06 ± 0.07 | 0.34 ± 0.42 | 0.12 ± 0.26 | 1.79 ± 1.52 | 0.42 ± 0.10 |

RL algorithm that regards out-of-distribution actions as unsafe and learns the reward critic using only safe state-action pairs. COptiDICE [31], a member of DIstribution Correction Estimation (DICE) family, is specifically designed for offline safe RL and directly estimates the stationary distribution correction of the optimal policy in terms of reward returns under safety constraints. CDT [37] is a DT-based algorithm that learns a policy conditioned on the target returns, as discussed in Section 2 as a preliminary. Finally, CCAC [22] is a recent proposed offline safe RL algorithm that models the relationship between state-action distributions and safety constraints and then leverages this relationship to regularize critics and policy learning. We use offline safe-RL algorithms as baselines because standard online approaches often violate safety constraints during training and optimize objectives that diverge from ours. Although some safe exploration algorithms share similar goals, they rely on strong assumptions—such as known and deterministic transition dynamics [51] or access to an emergency reset policy [44, 54]—that do not hold in our experimental setting.

**Implementation of PLS.** We use CDT [37] for offline policy learning as a constrained RCSL algorithm. The neural network configurations or hyperparameters for PLS are the same as the CDT used as a baseline. The key difference lies in how target returns are determined. In the baseline CDT, as a typical choice, we set the target reward return to the maximum reward return in the dataset and the target safety cost return to the threshold. In contrast, PLS employs GPs with radial basis function kernels to optimize the target returns for maximizing the reward under the safety constraint.

**Main results.** Table 1 summarizes our experimental results under a safety cost threshold of $b = 20$. Additional results, including Table 7 for $b = 40$, are provided in Appendix K. Notably, PLS is the only method that satisfies the safety constraint in every task. In contrast, every baseline algorithm violates the safety constraint in at least one task, which implies that a policy violating constraints could potentially persist in unsafe behavior in an actual environment. Moreover, PLS achieves the highest reward return in most tasks, which demonstrates its its superior overall performance in terms of reward and safety. In summary, while baseline methods suffer from either safety constraint violations or poor reward returns, PLS consistently delivers a balanced performance.

**Computational cost.** Although GPs are known to be computationally expensive, PLS only needs to optimize target returns in two dimensions, $z = (R, G)$. Because the amount of training data for the GPs is fairly small until convergence (see also Figure 3 in Appendix K), their computational overhead is not problematic. Consequently, the main source of computational cost in PLS stems from offline policy learning. Since PLS can adapt to multiple thresholds using a single policy by appropriately choosing target returns, it typically incurs lower overall computational cost than baseline algorithms (e.g., CPQ, COptiDICE), which require training a separate policy for each threshold.

**Safe exploration.** As shown in Figure 3 in Appendix K, PLS successfully ensures safety not only after convergence but also while exploring target returns, which is consistent with Theorem 2. In some cases, however, maintaining safety beyond the initial deployment can still pose a challenge in practice. Because our guarantee is probabilistic and constructing accurate GP models is not always feasible, a small number of unsafe deployments may occur.

## 8 Conclusion

We propose PLS as a solution to a longstanding goal in safe RL: achieving end-to-end safety from learning to operation. PLS consists of two key components: (1) offline policy learning via RCSL and (2) safe deployment that carefully optimizes target returns on which the pre-trained policy is conditioned. The relationship between target and actual returns is modeled using GPs, an approach justified by our theoretical analyses. We also provide theoretical guarantees on safety and near-optimality, and we empirically demonstrate the effectiveness of PLS in safe RL benchmark tasks.

**Limitations.** Our work has several limitations that open avenues for future research. First, while PLS guarantees near-optimal target returns, as established in Theorem 3, this does not directly translate into achieving a near-optimal policy. Second, our current framework does not update the policy network with new online data; that is, the policy is fixed after the initial offline training phase. Extending PLS to an offline-to-online setting where the policy continually learns while preserving safety guarantees is a crucial next step. Finally, while our experiments demonstrate strong performance, further evaluation in more complex and highly stochastic environments is needed to fully assess the practical robustness of our theoretical assumptions and the scalability of the approach.

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

# Appendix

## A Nomenclature

For readability, we present the main variables and functions below as a nomenclature table.

| Symbol | Description |
|---|---|
| $a_t$ | Action at time step $t$. |
| $b$ | Safety threshold for the cumulative safety cost. |
| $f(\cdot\|\cdot)$ | Conditional probability density function of the behavior return. |
| $g$ | Safety cost function, bounded in $[0,1]$. |
| $G$ | The target safety cost return. |
| $\widehat{G}(\tau)$ | The observed cumulative safety cost for a trajectory $\tau$. |
| $H$ | The fixed, finite length of each episode (horizon). |
| $J(\pi)$ | A pair of reward and safety returns, $(J_r(\pi), J_g(\pi))$. |
| $J_r(\pi)$ | The expected cumulative reward return for policy $\pi$. |
| $J_g(\pi)$ | The expected cumulative safety cost return for policy $\pi$. |
| $k(\cdot,\cdot)$ | Covariance (kernel) function for a Gaussian Process. |
| $L$ | Lipschitz constant. |
| $n$ | The number of trajectories in the offline dataset $\mathcal{D}$. |
| $N$ | The number of Gaussian Process (GP) observations or iterations. |
| $p_\theta(\cdot\|\cdot)$ | A parametric model of conditional action probability densities. |
| $P$ | State transition probability function, $P : \mathcal{S} \times \mathcal{A} \to \Delta(\mathcal{S})$. |
| $P_T$ | Transition kernel associated with $\langle P, r, g \rangle$. |
| $r$ | Reward function, bounded in $[0,1]$. |
| $R$ | The target reward return. |
| $\widehat{R}(\tau)$ | The observed cumulative reward for a trajectory $\tau$. |
| $s_t$ | State at time step $t$. |
| $x_t$ | The context at time step $t$. |
| $z$ | A pair of target returns, $z = (R, G)$. |
| | |
| $\alpha$ | Parameter balancing exploration and exploitation in Bayesian optimization. |
| $\beta$ | The return-conditioned behavior policy used to generate the offline dataset $\mathcal{D}$. |
| $\Delta$ | The allowed failure probability for safety guarantees (e.g., $1 - \Delta$). |
| $\varepsilon(z)$ | A small bias function in the theoretical analysis of returns. |
| $\eta_z$ | Initial coverage, defined as $f(z\|s_1)$. |
| $\mu(\cdot)$ | Mean function for a Gaussian Process. |
| $\pi$ | A policy that maps a context to an action distribution, $\pi : \mathcal{X} \to \Delta(\mathcal{A})$. |
| $\pi_z$ | A return-conditioned policy characterized by target returns $z$. |
| $\tau$ | A trajectory, $(\xi_1, \xi_2, ..., \xi_H)$. |
| $\theta$ | The parameters of the neural network policy model. |
| $\hat{\theta}$ | The maximum likelihood estimate of the model parameters $\theta$. |
| $\Xi$ | An episode or trajectory, $\{\xi_t\}_{t=1}^H$. |
| $\xi_t$ | A tuple at time $t$, defined as $(s_t, a_t, r_t, g_t)$. |
| $\zeta$ | A tolerance parameter for terminating safe exploration. |
| $\varpi$ | The number of trajectories used for sample approximations of returns for each GP update. |

## B Broader Impacts

We believe that our proposed approach PLS plays a significant role in enhancing the benefits associated with reinforcement learning while concurrently working to minimize any potential negative side effects. However, it must be acknowledged that any reinforcement learning algorithm, regardless of its design or intended purpose, is intrinsically susceptible to abuse, and we must remain cognizant of the fact that the fundamental concept underlying PLS can be manipulated or misused in ways that might ultimately render reinforcement learning systems less safe.

# C Pseudo Code of PLS

For completeness, we will present a pseudo code of our PLS.

---

**Algorithm 1** Provably Lifetime Safe Reinforcement Learning (PLS)

---

1: **Input:** Pre-collected dataset $\mathcal{D}$, safety threshold $b$, safe singleton set $\mathcal{Z}_0$, Lipschitz constant $L$
2:
3: *// Offline policy Learning (safe with probability of 1)*
4: Train a return-conditioned policy $\pi_{\boldsymbol{z}}$ from $\mathcal{D}$ via constrained RCSL
5:
6: *// Safe exploration (safe with high probability)*
7: Initialize $\mathcal{Y}_0$ with $\mathcal{Z}_0$
8: **for** $N = 1, \ldots, N_\dagger$ **do**
9: $\quad \mathcal{Y}_N \leftarrow \bigcup_{\boldsymbol{z} \in \mathcal{Y}_{N-1}} \{ \boldsymbol{z}' \in \mathcal{Z} \mid u_{g,N}(\boldsymbol{z}) + L \cdot d(\boldsymbol{z}, \boldsymbol{z}') \leq b \}$
10: $\quad e_N(\boldsymbol{z}) \leftarrow \left| \{ \boldsymbol{z}' \in \mathcal{Z} \setminus \mathcal{Y}_N \mid \ell_{g,N}(\boldsymbol{z}) - L \cdot d(\boldsymbol{z}, \boldsymbol{z}') \leq b \} \right|$
11: $\quad E_N \leftarrow \{ \boldsymbol{z} \in \mathcal{Y}_N : e_N(\boldsymbol{z}) > 0 \}$
12: $\quad \boldsymbol{z}_N \leftarrow \mathrm{argmax}_{\boldsymbol{z} \in E_N} \left( u_{\diamond,N}(\boldsymbol{z}) - \ell_{\diamond,N}(\boldsymbol{z}) \right)$
13: $\quad$ Update GPs using the reward and safety cost observations $J_r(\pi_{\boldsymbol{z}_N})$ and $J_g(\pi_{\boldsymbol{z}_N})$.
14: **end for**
15:
16: *// Reward maximization (safe with high probability)*
17: **for** $N = N_\dagger + 1, \ldots, N_\dagger + N_\sharp$ **do**
18: $\quad \mathcal{Y}_N \leftarrow \bigcup_{\boldsymbol{z} \in \mathcal{Y}_{N-1}} \{ \boldsymbol{z}' \in \mathcal{Z} \mid u_{g,N}(\boldsymbol{z}) + L \cdot d(\boldsymbol{z}, \boldsymbol{z}') \leq b \}$
19: $\quad \boldsymbol{z}_N \leftarrow \mathrm{argmax}_{\boldsymbol{z} \in \mathcal{Y}_N} u_{r,N}(\boldsymbol{z})$
20: $\quad$ Update GPs using the reward and safety cost observations $J_r(\pi_{\boldsymbol{z}_N})$ and $J_g(\pi_{\boldsymbol{z}_N})$.
21: **end for**
22:
23: *// Operation (safe with high probability)*
24: **while** true **do**
25: $\quad$ Continue to use $\boldsymbol{z}_N$ as target returns for long-term operation.
26: **end while**

---

# D Preliminaries of Theoretical Analyses

As a more general formulation of the problem, we define a multi-objective MDP characterized by $m$ reward functions, where $m$ is an arbitrary positive integer. Our theoretical analyses in the main paper are a specific case of $m = 2$ compared to those we will present in the following.

## D.1 Multi-objective Reinforcement Learning

Episodes are sequences of states, actions, and rewards $\Xi := \{(s_t, a_t, \boldsymbol{r}_t)\}_{t=1}^{H} \in (\mathcal{S} \times \mathcal{A} \times \mathbb{R}^m)^H$, where $H \geq 0$ is a time horizon and $m \geq 1$ is the number of reward dimensions. The $t$-th context $x_t$ of an episode refers to the partial history

$$x_t := (s_1, a_1, \boldsymbol{r}_1, \ldots, s_{t-1}, a_{t-1}, \boldsymbol{r}_{t-1}, s_t) \tag{13}$$

for $1 \leq t \leq H + 1$, where we let $s_{H+1} = \bot$ be a dummy state. Let $\mathcal{X}_t := (\mathcal{S} \times \mathcal{A} \times \mathbb{R}^m)^{t-1} \times \mathcal{S}$ be the set of all $t$-th contexts and $\mathcal{X} := \bigcup_{t=1}^{H} \mathcal{X}_t$ be the sets of all contexts at steps $1 \leq t \leq H$.

With a fixed initial state $s_1$ and a transition kernel $P_T : \mathcal{S} \times \mathcal{A} \rightarrow \Delta(\mathbb{R}^m \times \mathcal{S})$, we consider the Markov decision process (MDP) $\mathcal{M} = (\mathcal{S}, \mathcal{A}, H, s_1, P_T)$.[3] Under $\mathcal{M}$, every (context-dependent) policy $\pi : \mathcal{X} \rightarrow \Delta(\mathcal{A})$ identifies a probability distribution $\mathbb{P}^\pi$ on $\Xi$ such that $a_t \sim \pi(x_t)$ and $(\boldsymbol{r}_t, s_{t+1}) \sim P_T(s_t, a_t)$ for all $t \geq 1$.

**Assumption 8** (Bounded reward). For any policies $\pi$, we have $\mathbb{P}^\pi$-almost surely $0 \leq \boldsymbol{r}_{t,j} \leq 1$ for $1 \leq t \leq H$ and $1 \leq j \leq m$.

---

[3] Our analysis can be easily extended to $s_1$ being stochastic.

**Assumption 9** (Near-deterministic transition). There exist deterministic maps $\hat{\boldsymbol{r}}(\cdot, \cdot)$, $\hat{s}'(\cdot, \cdot)$ and small constants $\epsilon_r, \epsilon_s, \delta \geq 0$ such that, if $(\boldsymbol{r}, s') \sim P_T(s, a)$,

1. the reward density $p_r(\boldsymbol{r}'|s, a) := \frac{\mathrm{d}}{\mathrm{d}\boldsymbol{r}'} P_T\{\boldsymbol{r} \leq \boldsymbol{r}'|s, a\}^4$ is well-defined and bounded by $\epsilon_r$ outside the $\delta$-neighborhood of $\hat{\boldsymbol{r}}(s, a)$, i.e., $\sup_{\boldsymbol{r}:\|\boldsymbol{r}-\hat{\boldsymbol{r}}(s,a)\|_\infty > \delta} p_r(\boldsymbol{r}|s, a) \leq \epsilon_r$, and

2. the successor state $s'$ coincides with $\hat{s}'(s, a)$ with probability of at least $1 - \epsilon_s$,

for all $s \in \mathcal{S}$ and $a \in \mathcal{A}$.

Let $\beta : \mathcal{X} \to \Delta(\mathcal{A})$ be a behavior policy and $\mathcal{D} := \{\Xi^{(i)}\}_{i=1}^n \sim (\mathbb{P}^\beta)^n$ be a collection of $n$ i.i.d. copies of episodes generated by $\beta$.

**Assumption 10** (Reward-independent behavior). The behavior action distribution $\beta(x_t)$, $x_t \in \mathcal{X}$, is conditionally independent of the past rewards $\{\boldsymbol{r}_h\}_{h=1}^{t-1}$ given the past states and actions $x_t \setminus \{\boldsymbol{r}_h\}_{h=1}^{t-1}$.

Let $\boldsymbol{J}(\pi)$ denote the multi-dimensional policy value of $\pi$,

$$\boldsymbol{J}(\pi) = (J_1(\pi), \ldots, J_m(\pi)) := \mathbb{E}^\pi[\widehat{\boldsymbol{R}}] \in \mathbb{R}^m, \tag{14}$$

where $\widehat{\boldsymbol{R}} := \sum_{t=1}^H \boldsymbol{r}_t$ denotes the return of episode and the superscript $\pi$ of $\mathbb{E}^\pi$ signifies the dependency on $\mathbb{P}^\pi$.

The aforementioned setting leads to constrained RL problems where a policy aims to maximize one dimension of the policy value $J_1(\pi)$ as much as possible while controlling the other dimensions to satisfy constraints $J_k(\pi) \leq b_k$ with certain threshold $b_k \in \mathbb{R}$, for $2 \leq k \leq m$. More specifically, $\boldsymbol{r}_1$ and $\boldsymbol{r}_2$ respectively correspond to $r$ and $g$ in the main paper.

## D.2 Return-conditioned supervised learning

Return-conditioned supervised learning (RCSL) is a methodology of offline reinforcement learning that aims at estimating the return-conditioned behavior (RCB) policy $\beta_{\boldsymbol{R}}(a|x) := \mathbb{P}^\beta(a_t = a|x_t = x, \widehat{\boldsymbol{R}} = \boldsymbol{R})$, the action distribution conditioned on the return $\widehat{\boldsymbol{R}} = \boldsymbol{R} \in [0, H]^m$ as well as the context $x_t = x \in \mathcal{X}$. According to the Bayes' rule, the RCB policy $\beta_{\boldsymbol{R}} : \mathcal{X} \to \Delta(\mathcal{A})$ is written as the importance-weighted behavior policy

$$\mathrm{d}\beta_{\boldsymbol{R}}(a \mid x) = \frac{f(\boldsymbol{R} \mid x, a)}{f(\boldsymbol{R} \mid x)} \mathrm{d}\beta(a \mid x), \tag{15}$$

where $f(\boldsymbol{R} \mid x) := \frac{\mathrm{d}}{\mathrm{d}\boldsymbol{R}} \mathbb{P}^\beta(\widehat{\boldsymbol{R}} \leq \boldsymbol{R} \mid x_t = x)$ and $f(\boldsymbol{R} \mid x, a) := \frac{\mathrm{d}}{\mathrm{d}\boldsymbol{R}} \mathbb{P}^\beta(\widehat{\boldsymbol{R}} \leq \boldsymbol{R} \mid x_t = x, a_t = a)$ respectively denote the conditional probability density functions of the behavior return.[5]

Return-based importance weighting (15) favors the actions that led to the target return $\boldsymbol{R}$ over those that did not. Hence, intuitively, it is expected that $\beta_{\boldsymbol{R}}$ achieves

$$J(\beta_{\boldsymbol{R}}) \approx \boldsymbol{R}. \tag{16}$$

This is the case under suitable assumptions. Thus we can solve multi-objective reinforcement learning with RCSL by setting $\boldsymbol{R}$ to a desired value.

We assume the following conditions on $f(\boldsymbol{R} \mid x)$, with $\boldsymbol{R}$ fixed to a value of interest.

**Assumption 11** (Initial coverage). $\eta_{\boldsymbol{R}} := f(\boldsymbol{R} \mid s_1) > 0$.

**Assumption 12** (Boundedness). $C_{\boldsymbol{R}} := \sup_{x \in \mathcal{X}} f(\boldsymbol{R} \mid x) < \infty$.

**Assumption 13** (Continuity). $c_{\boldsymbol{R}}(\delta) := \sup_{\boldsymbol{R}':\|\boldsymbol{R}'-\boldsymbol{R}\|_\infty \leq 2\delta, x \in \mathcal{X}} |f(\boldsymbol{R}'|x) - f(\boldsymbol{R}|x)| < \infty$ is small.

---

[4]We abuse the notation $\boldsymbol{r} \leq \boldsymbol{r}'$ for $\boldsymbol{r}, \boldsymbol{r}' \in \mathbb{R}^m$ to imply the multi-dimensional inequality, i.e., $\boldsymbol{r}_j \leq \boldsymbol{r}'_j$ for all $1 \leq j \leq m$.

[5]Strictly speaking, the RHS of (15) may be ill-defined for some $x \in \mathcal{X}$ and $a \in \mathcal{A}$ if either $f(\boldsymbol{R}|x)$ or $f(\boldsymbol{R}|x, a)$ are ill-defined, or $f(\boldsymbol{R}|x) = 0$. However, it is sufficient for our analysis to impose (15) on $\beta_{\boldsymbol{R}}$ only if the RHS is well-defined.

## D.3 Decision transformers

Decision transformer (DT) is an implementation of RCSL. More specifically, it is seen as a regularized maximum likelihood estimation (MLE) method

$$\hat{\theta} = \operatorname*{argmin}_{\theta \in \Theta} \left\{ -\frac{1}{nH} \sum_{i=1}^{n} \sum_{t=1}^{H} \ln p_\theta(a_t^{(i)} \mid x_t^{(i)}, \widehat{\boldsymbol{R}}^{(i)}) + \Phi(\theta) \right\}, \tag{17}$$

where $\mathcal{P} := \{p_\theta(a \mid x, \boldsymbol{R})\}_{\theta \in \Theta}$ is a parametric model of conditional probability densities, typically constructed with the transformer architecture, and $\Phi(\theta) \geq 0$ is a penalty term representing inductive biases, both explicit and implicit, in the procedure of parameter optimization. Here, $a_t^{(i)}$, $x_t^{(i)}$ and $\widehat{\boldsymbol{R}}^{(i)}$ are the $t$-th action, the $t$-th context, and the return of the $i$-th episode $\Xi^{(i)} \in \mathcal{D}$, respectively. The output of decision transformer is then given by $\pi_{\hat{\theta}, \boldsymbol{R}}$, where $\pi_{\theta, \boldsymbol{R}}$ denotes the policy associated with $p_\theta(\cdot \mid \cdot, \boldsymbol{R})$. Note that the original DT is for a single-dimensional reward function, we presented (17) by extending it to multi-dimensional settings.

We introduce some notation and conditions on the probabilistic model $\mathcal{P}$ and the penalty $\Phi$. Let us define a regularized risk of $\theta$ relative to $\beta_{\boldsymbol{R}}$ by

$$\mathcal{R}_\Phi(\theta) := \underbrace{\mathbb{E}_{t \sim \mathrm{Unif}[H]}^\beta \left[ D_{\mathrm{KL}}(\beta_{\widehat{\boldsymbol{R}}}(x_t) \| \pi_{\theta, \widehat{\boldsymbol{R}}}(x_t)) \right]}_{\text{dissimilarity of } \beta_{\boldsymbol{R}} \text{ and } \pi_{\theta, R} \text{ in expectation}} + \Phi(\theta), \tag{18}$$

where $D_{\mathrm{KL}}(\cdot \| \cdot)$ denotes the Kullback–Leibler divergence.

**Assumption 14** (Soft realizability). $\epsilon_{\mathcal{P}, \Phi} := \min_{\theta \in \Theta} \mathcal{R}_\Phi(\theta) < \infty$ is small.

**Remark 2.** Assumption 14 is a relaxation of a standard realizability condition. That is, we have $\epsilon_{\mathcal{P}, \Phi} = 0$ if $\beta_{\boldsymbol{R}}$ is realizable in $\mathcal{P}$ without penalty, i.e., there exists $\theta_0 \in \Theta$ such that $\pi_{\theta_0, \boldsymbol{R}} = \beta_{\boldsymbol{R}}$ and $\Phi(\theta_0) = 0$.

**Assumption 15** (Regularity). The following conditions are met.

i) $\Theta$ is a compact subset of $\mathbb{R}^d$, $d \geq 1$.

ii) $\mathcal{R}_\Phi(\theta)$ admits a unique minimizer $\theta^*$ in the interior set $\Theta^\circ$.

iii) $\mathcal{R}_\Phi(\theta)$ is twice differentiable at $\theta^*$ with Hessian $\mathcal{I}_{\theta^*} := \nabla_\theta^2 \mathcal{R}_\Phi(\theta^*) \succ 0$.

iv) The one-sample stochastic gradient $\psi_\theta(a|x, \boldsymbol{R}) := \nabla_\theta\{-\ln p_\theta(a|x, \boldsymbol{R}) + \Phi(\theta)\}$ is locally bounded in expectation as

$$\mathbb{E}_{t \sim \mathrm{Unif}[H]}^\beta \left[ \sup_{\theta \in \Theta_b} \left\| \psi_\theta(a_t|x_t, \widehat{\boldsymbol{R}}) \right\|_2^2 \right] < \infty \tag{19}$$

for every sufficiently small ball $\Theta_b$ in $\Theta$.

v) $\hat{\theta} \in \Theta^\circ$ almost surely.

**Remark 3.** At first glance, ii) the unique existence of $\theta^*$ and iii) the positive definiteness of the Hessian seem restrictive for over-parametrized models, including transformers. However, we note that these conditions may be enforced by adding a tiny, strongly convex penalty to $\Phi(\theta)$.

**Remark 4.** Similarly, v) $\hat{\theta} \in \Theta^\circ$ can be also enforced by adding a barrier function such as $\Phi(\theta) = K\phi_{\mathrm{hinge}}^2(\mathrm{dist}(\theta, \mathbb{R}^d \setminus \Theta)/h)$, where $h > 0$ and $K < \infty$ are respectively suitably small and large constants, $\mathrm{dist}(\theta, E) := \inf_{\theta' \in E} \|\theta - \theta'\|_2$, and $\phi_{\mathrm{hinge}}(t) := \max\{0, 1 - t\}$.

# E    Error analysis

Our goal here is to understand when and how closely the output of decision transformer, $\pi_{\hat{\theta}, \boldsymbol{R}}$, achieves the target return, $\boldsymbol{R}$. The following theorem summarizes our theoretical results, answering the above question.

**Theorem 4.** *Under the assumptions of Theorems 5 to 7, we have*

$$\left\| \boldsymbol{J}(\pi_{\hat{\theta},\boldsymbol{R}}) - \boldsymbol{R} - \frac{H^2}{\sqrt{n}}\mathcal{F}(\boldsymbol{R}) \right\|_\infty \leq \varepsilon(\boldsymbol{R}) + o_P\left(\frac{1}{\sqrt{n}}\right), \tag{20}$$

*where $\mathcal{F} : [0, H]^m \to \mathbb{R}^m$ is a sample path of a Gaussian process with mean zero and $\varepsilon(\boldsymbol{R}) := \frac{2\bar{C}_{\boldsymbol{R}}(H^2\epsilon+\delta)+H^2 c_{\boldsymbol{R}}(\delta)}{\eta_{\boldsymbol{R}}} + H^2\sqrt{\frac{\epsilon_{\mathcal{P},\Phi}}{2}}$ is a small bias function, where $\bar{C}_{\boldsymbol{R}} = \max\{C_{\boldsymbol{R}}, 1\}$ and $\epsilon = \epsilon_r + \epsilon_s$. Here, $o_P(\cdot)$ is the probabilistic small-o notation, i.e, $b_n = o_P(a_n)$ signifies $\lim_{n\to\infty}\mathbb{P}\{|b_n/a_n| > \epsilon\} = 0$ for all $\epsilon > 0$.*

**Remark 5.** Theorem 1 in the main paper is a special case of Theorem 4 of $m = 2$, which is presented in a slightly informal manner.

To derive Theorem 4, we consider the bias-variance decomposition

$$J(\pi_{\hat{\theta},\boldsymbol{R}}) - \boldsymbol{R} = \underbrace{J(\beta_{\boldsymbol{R}}) - \boldsymbol{R}}_{\text{bias of RCSL}} + \underbrace{J(\pi_{\theta^*,\boldsymbol{R}}) - J(\beta_{\boldsymbol{R}})}_{\text{bias of MLE}} + \underbrace{J(\pi_{\hat{\theta},\boldsymbol{R}}) - J(\pi_{\theta^*,\boldsymbol{R}})}_{\text{variance of MLE}} \tag{21}$$

and evaluate each term in RHS with Theorems 5 to 7, respectively, through Appendices E.1 and E.2.

### E.1  Bias of RCSL

The following theorem gives an upper bound on the first bias term, showing that it is negligible under suitable conditions, such as the near-determinism of the transition and the regularity of the return density. The proof is deferred to Appendix F.

**Theorem 5.** *Suppose Assumptions 8 to 13 hold. Then,*

$$\|J(\beta_{\boldsymbol{R}}) - \boldsymbol{R}\|_\infty \leq \frac{2\bar{C}_{\boldsymbol{R}}\left(H^2\epsilon + \delta\right) + H^2 c_{\boldsymbol{R}}(\delta)}{\eta_{\boldsymbol{R}}}, \tag{22}$$

*where $\epsilon := \epsilon_r + \epsilon_s$ and $\bar{C}_{\boldsymbol{R}} := \max\{C_{\boldsymbol{R}}, 1\}$.*

A few remarks follow in order. First, we compare our result to previous one.

**Remark 6.** Theorem 5 can be considered as a complementary extension of the previous result [9]. In particular, our result is applicable when the return density $f(\boldsymbol{R}|s_1)$ is bounded away from 0 and $\infty$, while Theorem 1 of [9] is not. On the contrary, Theorem 1 of [9] is applicable when there is a nonzero probability of exactly $\boldsymbol{R} = \widehat{\boldsymbol{R}}$, while our result is not since $f(\boldsymbol{R}|s_1) = \infty$.

**Remark 7.** Our result also extends Theorem 1 in Brandfonbrener et al. [9] in allowing the transition kernel $P_T$ to include small additive noises in the reward, i.e., $\delta > 0$.

Below is a generalization of (22) that is useful to understand what constitutes the upper bound.

**Remark 8.** Taking a closer look at the proof of Theorem 5, we can conclude

$$\|J(\beta_{\boldsymbol{R}}) - \boldsymbol{R}\|_\infty \leq \frac{2\bar{C}_{\boldsymbol{R}}\left(H^2\epsilon + \delta_H\right) + \sum_{t=1}^{H-1} H c_{\boldsymbol{R}}(\delta_t)}{\eta_{\boldsymbol{R}}}, \tag{23}$$

where $\delta_t$ is the additive noise tolerance specific to the $t$-th transition. In other words, the contributions of these additive errors to the bias of RCSL depends largely on whether they are in the terminal step ($t = H$) or not.

If we have Assumption 9 with $\delta = 0$, Assumption 13 is automatically satisfied with $c_{\boldsymbol{R}}(0) = 0$ and Assumption 10 is unnecessary, resulting in the following rather simplified corollary.

**Corollary 1.** *Suppose Assumptions 8, 9, 11 and 12 hold with $\delta = 0$. Then,*

$$\|J(\beta_{\boldsymbol{R}}) - \boldsymbol{R}\|_\infty \leq \frac{2\bar{C}_{\boldsymbol{R}}H^2\epsilon}{\eta_{\boldsymbol{R}}}. \tag{24}$$

Besides, Assumption 12 can be replaced with a stronger variant of Assumption 13.

**Corollary 2.** *Suppose Assumptions 8 to 11 hold. Also assume the Hölder continuity of $f(\cdot|x)$,*

$$|f(\boldsymbol{R}'|x) - f(\boldsymbol{R}|x)| \le K \|\boldsymbol{R}' - \boldsymbol{R}\|_{\infty}^{\omega}, \quad \boldsymbol{R}, \boldsymbol{R}' \in [0, H], \quad x \in \mathcal{X}. \tag{25}$$

*Then,*

$$\|J(\beta_{\boldsymbol{R}}) - \boldsymbol{R}\|_{\infty} \le \frac{2(K+1)}{\eta_{\boldsymbol{R}}} \left\{ H^2(\epsilon + \delta^{\omega}) + \delta \right\}. \tag{26}$$

*Proof.* It directly follows from that $C_{\boldsymbol{R}} \le K + 1$ and $c_{\boldsymbol{R}}(\delta) \le K(2\delta)^{\omega} \le 2K\delta^{\omega}$. See Lemma 3 for the argument on bounding $C_{\boldsymbol{R}}$. $\qquad\square$

### E.2 Bias and variance of MLE

The following theorem shows that the bias of MLE in (21) is negligible if a mild realizability condition is met. The proof is deferred to Appendix G.

**Theorem 6.** *Suppose Assumption 14 holds. Then,*

$$\|J(\pi_{\theta^*, \boldsymbol{R}}) - J(\beta_{\boldsymbol{R}})\|_{\infty} \le H^2 \sqrt{\frac{\epsilon_{\mathcal{P}, \Phi}}{2}}. \tag{27}$$

Moreover, the following theorem characterizes the asymptotic distribution of the variance of MLE in (21). The proofs are deferred to Appendix H. Let us introduce the gradient covariance matrix

$$\mathcal{V}_{\theta} := \mathbb{E}_{t \sim \mathrm{Unif}[H]}^{\beta} \left[ \psi_{\theta}(a_t|x_t, \widehat{\boldsymbol{R}}) \psi_{\theta}(a_t|x_t, \widehat{\boldsymbol{R}})^{\top} \right] \in \mathbb{R}^{d \times d} \tag{28}$$

and the normalized policy Jacobian

$$U_{\theta}(\boldsymbol{R}) := \frac{1}{H} \mathbb{E}_{t \sim \mathrm{Unif}[H]}^{\pi_{\theta}, \boldsymbol{R}} \left[ Q^{\pi_{\theta}, \boldsymbol{R}}(x_t, a_t) \nabla_{\theta} \ln p_{\theta}(a_t|x_t, \boldsymbol{R})^{\top} \right] \in \mathbb{R}^{m \times d}, \tag{29}$$

where $Q^{\pi}(x, a) := \mathbb{E}^{\pi}[\widehat{\boldsymbol{R}}|x_t = x, a_t = a] \in \mathbb{R}^m$ is the $m$-dimensional action value function.

**Theorem 7.** *Suppose Assumption 15 holds. Then, we have*

$$\left\{ \frac{\sqrt{n}}{H^2} \left[ J_j(\pi_{\hat{\theta}, \boldsymbol{R}}) - J_j(\pi_{\theta^*, \boldsymbol{R}}) \right] \right\}_{j \in [m], \boldsymbol{R} \in [0, H]^m} \rightsquigarrow \mathsf{GP}(0, \boldsymbol{k}) \tag{30}$$

*in the limit of $n \to \infty$, where $\boldsymbol{k}: [0, H]^m \times [0, H]^m \to \mathbb{R}^{m \times m}$ is the covariance function given by*

$$\boldsymbol{k}(\boldsymbol{R}, \boldsymbol{R}') := U_{\theta^*}(\boldsymbol{R}) \mathcal{I}_{\theta^*}^{-1} \mathcal{V}_{\theta^*} \mathcal{I}_{\theta^*}^{-1} U_{\theta^*}(\boldsymbol{R}')^{\top}. \tag{31}$$

**Remark 9.** The differentiability of sample paths of the limit process $\mathcal{F}(\cdot) \sim \mathsf{GP}(0, \boldsymbol{k})$ is known to be (roughly) the same as the differentiability of the covariance function $\boldsymbol{k}(\cdot, \cdot)$ (Corollary 1 in [13]), which, according to (31), is governed by that of $U_{\theta^*}(\cdot)$. In other words, $\mathcal{F}(\cdot)$ is smooth if $U_{\theta^*}(\cdot)$ is smooth. With a straightforward calculation, one can further see that $U_{\theta^*}(\cdot)$ is smooth if, under some mild regularity conditions, the probabilistic model $\mathcal{P}$ is smooth in terms of the associated policy $\pi_{\theta^*, \boldsymbol{R}}$ and the gradient $\nabla_{\theta} \ln p_{\theta}(a_t|x_t, \boldsymbol{R})|_{\theta=\theta^*}$ as functions of the target return $\boldsymbol{R}$.

## F Proof of Theorem 5

Consider the weighted error function given by

$$\phi(x_t) := f(\boldsymbol{R}|x_t) \|\boldsymbol{V}(x_t) - \hat{\boldsymbol{V}}(x_t)\|_{\infty}, \tag{32}$$

where $\boldsymbol{V}(x_t) := \mathbb{E}^{\beta_{\boldsymbol{R}}}[\sum_{h=t}^{H} \boldsymbol{r}_h|x_t]$ is the value function of $\beta_{\boldsymbol{R}}$ and $\hat{\boldsymbol{V}}(x_t) := \boldsymbol{R} - \sum_{h=1}^{t-1} \boldsymbol{r}_h$ is the target value function. It suffices for the proof of Theorem 5 to establish a suitable bound on $\phi(x_1)$ since, by Assumption 11,

$$\|\boldsymbol{J}(\beta_{\boldsymbol{R}}) - \boldsymbol{R}\|_{\infty} = \frac{\phi(x_1)}{f(\boldsymbol{R}|x_1)} = \frac{\phi(x_1)}{\eta_{\boldsymbol{R}}}. \tag{33}$$

To this end, we will make use of $\hat{P}_T : \mathcal{S} \times \mathcal{A} \to \Delta(\mathbb{R}^m \times \mathcal{S})$, the near-deterministic component of $P_T$ such that

$$\mathrm{d}\hat{P}_T(\boldsymbol{r}, s'|s, a) = \frac{\mathbb{I}\{(\boldsymbol{r}, s') \in \hat{\mathcal{T}}(s, a)\}}{P_T(\hat{\mathcal{T}}(s, a)|s, a)} \mathrm{d}P_T(\boldsymbol{r}, s'|s, a), \tag{34}$$

where $\hat{\mathcal{T}}(s, a) = B_\infty(\hat{\boldsymbol{r}}(s, a), \delta) \times \{\hat{s}'(s, a)\} \subset \mathbb{R}^m \times \mathcal{S}$ is the image of the near-deterministic transition and $B_\infty(\boldsymbol{r}, \delta) := \{\boldsymbol{r}' \in \mathbb{R}^m : \|\boldsymbol{r}' - \boldsymbol{r}\|_\infty \leq \delta\}$ is the $\ell^\infty$-ball centered at $\boldsymbol{r}$ with radius $\delta$. Let also $\hat{\mathbb{P}}, \hat{\mathbb{E}}, \hat{\mathbb{P}}^\pi, \hat{\mathbb{E}}^\pi$ be probability distributions and expectation operators identical to $\mathbb{P}, \mathbb{E}, \mathbb{P}^\pi, \mathbb{E}^\pi$, respectively, except that the transition kernel $P_T$ is replaced with $\hat{P}_T$ under the hood.

Now, for $1 \leq t \leq H - 1$, we can bound $\phi(x_t)$ in terms of $\phi(x_{t+1})$.

**Lemma 1.** *Suppose Assumptions 8 to 10, 12 and 13 hold. Then, for all $x_t \in \mathcal{X}_t$ with $1 \leq t \leq H - 1$, we have*

$$\phi(x_t) \leq \hat{\mathbb{E}}^\beta \left[\phi(x_{t+1}) \,|\, x_t\right] + Hc_{\boldsymbol{R}}(\delta) + 2\epsilon HC_{\boldsymbol{R}}. \tag{35}$$

*Proof.* Let $\hat{f}(\boldsymbol{R}|x_t, a_t) := \hat{\mathbb{E}}[f(\boldsymbol{R}|x_{t+1})|x_t, a_t]$. Note that $f(\boldsymbol{R}'|x_t, a_t) = \mathbb{E}[f(\boldsymbol{R}'|x_{t+1})|x_t, a_t]$ is well-defined for all $x_t \in \mathcal{X}_t$ and $a_t \in \mathcal{A}$ by Assumptions 12 and 13. Thus, the claim follows from

$$\begin{aligned}
\phi(x_t) &= f(\boldsymbol{R}|x_t)\left\|\boldsymbol{V}(x_t) - \hat{\boldsymbol{V}}(x_t)\right\|_\infty \\
&\overset{(a)}{\leq} f(\boldsymbol{R}|x_t) \int \left\|\boldsymbol{V}(x_{t+1}) - \hat{\boldsymbol{V}}(x_{t+1})\right\|_\infty \mathrm{d}\mathbb{P}^{\beta_{\boldsymbol{R}}}(a_t, \boldsymbol{r}_t, s_{t+1}|x_t) \\
&\overset{(b)}{\leq} f(\boldsymbol{R}|x_t) \int \left\|\boldsymbol{V}(x_{t+1}) - \hat{\boldsymbol{V}}(x_{t+1})\right\|_\infty \mathrm{d}\hat{\mathbb{P}}^{\beta_{\boldsymbol{R}}}(a_t, \boldsymbol{r}_t, s_{t+1}|x_t) + \epsilon HC_{\boldsymbol{R}} \\
&\overset{(c)}{=} \int \left\|\boldsymbol{V}(x_{t+1}) - \hat{\boldsymbol{V}}(x_{t+1})\right\|_\infty f(\boldsymbol{R}|x_t, a_t)\mathrm{d}\hat{\mathbb{P}}^\beta(a_t, \boldsymbol{r}_t, s_{t+1}|x_t) + \epsilon HC_{\boldsymbol{R}} \\
&\overset{(d)}{\leq} \int \left\|\boldsymbol{V}(x_{t+1}) - \hat{\boldsymbol{V}}(x_{t+1})\right\|_\infty \hat{f}(\boldsymbol{R}|x_t, a_t)\mathrm{d}\hat{\mathbb{P}}^\beta(a_t, \boldsymbol{r}_t, s_{t+1}|x_t) + 2\epsilon HC_{\boldsymbol{R}} \\
&\overset{(e)}{\leq} \int \left\|\boldsymbol{V}(x_{t+1}) - \hat{\boldsymbol{V}}(x_{t+1})\right\|_\infty f(\boldsymbol{R}|x_{t+1})\mathrm{d}\hat{\mathbb{P}}^\beta(a_t, \boldsymbol{r}_t, s_{t+1}|x_t) + Hc_{\boldsymbol{R}}(\delta) + 2\epsilon HC_{\boldsymbol{R}} \\
&= \int \phi(x_{t+1})\mathrm{d}\hat{\mathbb{P}}^\beta(a_t, \boldsymbol{r}_t, s_{t+1}|x_t) + Hc_{\boldsymbol{R}}(\delta) + 2\epsilon HC_{\boldsymbol{R}},
\end{aligned}$$

where (a) is shown by Jensen's inequality with $\boldsymbol{V}(x_t) - \hat{\boldsymbol{V}}(x_t) = \mathbb{E}^{\beta_{\boldsymbol{R}}}[\boldsymbol{V}(x_{t+1}) - \hat{\boldsymbol{V}}(x_{t+1})|x_t]$, (b) shown by Assumption 8 implying $\|\boldsymbol{V}(x) - \hat{\boldsymbol{V}}(x)\|_\infty \leq H$, Assumption 12 and Lemma 4 and, (c) shown by (15), (d) shown by Assumption 12 and evaluating $\hat{f}(\boldsymbol{R}|x_t, a_t) - f(\boldsymbol{R}|x_t, a_t) = \int f(\boldsymbol{R}|x_{t+1})\mathrm{d}\{\hat{P}_T - P_T\}(\boldsymbol{r}_t, s_{t+1}|s_t, a_t)$ with Lemma 4, and (e) shown by Lemma 5. $\square$

Finally, the proof of Theorem 5 is concluded by dealing with the boundary term $\phi(x_H)$.

**Lemma 2.** *Suppose Assumptions 8 to 10 and 13 hold. For all $x_H \in \mathcal{X}_H$, we have*

$$\phi(x_H) \leq 2\epsilon H\bar{C}_{\boldsymbol{R}} + 2\delta C_{\boldsymbol{R}}. \tag{36}$$

*Proof.* Similarly as the proof of Lemma 1, we have

$$\phi(x_H) \leq \int \left\|\boldsymbol{V}(x_{H+1}) - \hat{\boldsymbol{V}}(x_{H+1})\right\|_\infty f(\boldsymbol{R}|x_H)\mathrm{d}\hat{\mathbb{P}}^{\beta_{\boldsymbol{R}}}(a_H, \boldsymbol{r}_H|x_H) + \epsilon HC_{\boldsymbol{R}}.$$

We evaluate the RHS above by separating the domain of integral into two: i) where $a_H \in \mathcal{A}_{\mathrm{dtm}} := \{a \in \mathcal{A} : \|\hat{\boldsymbol{r}}(s_H, a_H) - \hat{\boldsymbol{V}}(x_H)\|_\infty \leq \delta\}$ and ii) where $a_H \notin \mathcal{A}_{\mathrm{dtm}}$. For the case i), we have

$$\left\|\boldsymbol{V}(x_{H+1}) - \hat{\boldsymbol{V}}(x_{H+1})\right\|_\infty \leq \|\boldsymbol{r}_H - \hat{\boldsymbol{r}}(s_H, a_H)\|_\infty + \left\|\hat{\boldsymbol{r}}(s_H, a_H) - \hat{\boldsymbol{V}}(x_H)\right\|_\infty \leq 2\delta$$

and therefore, by Assumption 12, the integral restricted to $\mathcal{A}_{\text{dtm}}$ is bounded with $2\delta C_{\boldsymbol{R}}$. For the case ii), note that $f(\boldsymbol{R}|x_H, a_H) = p_r(\hat{\boldsymbol{V}}(x_H)|s_H, a_H)$ is well-defined by Assumption 9 with $\|\hat{\boldsymbol{V}}(x_H) - \hat{\boldsymbol{r}}(s_H, a_H)\|_\infty > \delta$. Thus, we have

$$\int_{a_H \notin \mathcal{A}_{\text{dtm}}} \left\|\boldsymbol{V}(x_{H+1}) - \hat{\boldsymbol{V}}(x_{H+1})\right\|_\infty f(\boldsymbol{R}|x_H)\mathrm{d}\hat{\mathbb{P}}^{\beta_{\boldsymbol{R}}}(a_H, \boldsymbol{r}_H|x_H)$$

$$\overset{(a)}{=} \int_{a_H \notin \mathcal{A}_{\text{dtm}}} \left\|\boldsymbol{V}(x_{H+1}) - \hat{\boldsymbol{V}}(x_{H+1})\right\|_\infty f(\boldsymbol{R}|x_H, a_H)\mathrm{d}\hat{\mathbb{P}}^\beta(a_H, \boldsymbol{r}_H|x_H)$$

$$= \int_{a_H \notin \mathcal{A}_{\text{dtm}}} \left\|\boldsymbol{V}(x_{H+1}) - \hat{\boldsymbol{V}}(x_{H+1})\right\|_\infty p_r(\hat{\boldsymbol{V}}(x_H)|s_H, a_H)\mathrm{d}\hat{\mathbb{P}}^\beta(a_H, \boldsymbol{r}_H|x_H)$$

$$\overset{(b)}{\le} H\epsilon_r \le H\epsilon,$$

where (a) follows from (15) and (b) from Assumption 9. Combining both cases, we arrive at the desired result. $\qquad\square$

## G   Proof of Theorem 6

For simplicity, let $\pi_{\boldsymbol{R}}^* := \pi_{\theta^*, \boldsymbol{R}}$. By the performance difference lemma (Lemma 6), we have

$$\boldsymbol{J}(\pi_{\boldsymbol{R}}^*) - \boldsymbol{J}(\beta_{\boldsymbol{R}}) = \sum_{t=1}^{H} \mathbb{E}^{\beta_{\boldsymbol{R}}} \left[\boldsymbol{Q}^{\pi_{\boldsymbol{R}}^*}(x_t, \pi_{\boldsymbol{R}}^*(x_t)) - \boldsymbol{Q}^{\pi_{\boldsymbol{R}}^*}(x_t, \beta_{\boldsymbol{R}}(x_t))\right], \tag{37}$$

where RHS is further bounded by

$$\overset{(a)}{\le} H \sum_{t=1}^{H} \mathbb{E}^{\beta_{\boldsymbol{R}}} \left[\|\pi_{\boldsymbol{R}}^*(x_t) - \beta_{\boldsymbol{R}}(x_t)\|_{\text{TV}}\right] \tag{38}$$

$$= H^2 \mathbb{E}_{t \sim \text{Unif}[H]}^{\beta_{\boldsymbol{R}}} \left[\|\pi_{\boldsymbol{R}}^*(x_t) - \beta_{\boldsymbol{R}}(x_t)\|_{\text{TV}}\right] \tag{39}$$

$$\overset{(b)}{\le} H^2 \mathbb{E}_{t \sim \text{Unif}[H]}^{\beta_{\boldsymbol{R}}} \left[\sqrt{\frac{1}{2} D_{\text{KL}}(\beta_{\boldsymbol{R}}(x_t)\|\pi_{\boldsymbol{R}}^*(x_t))}\right] \tag{40}$$

$$\overset{(c)}{\le} H^2 \sqrt{\frac{1}{2} \mathbb{E}_{t \sim \text{Unif}[H]}^{\beta_{\boldsymbol{R}}} [D_{\text{KL}}(\beta_{\boldsymbol{R}}(x_t)\|\pi_{\boldsymbol{R}}^*(x_t))]} \tag{41}$$

$$\overset{(d)}{=} H^2 \sqrt{\frac{1}{2}\epsilon_{\mathcal{P}, \Phi}}. \tag{42}$$

Here, (a) is owing to the boundedness of the Q-function $0 \le \boldsymbol{Q}^\pi(x, a) \le H$, (b) is to Pinsker's inequality, (c) is to Jensen's, and (d) is to Assumption 14.

## H   Proof of Theorem 7

Note that $\hat{\theta}$ is the M-estimator [52] associated with the criterion function

$$M_\theta(a|x, R) := \ln \frac{p_\theta(a|x, R)}{p_{\theta^*}(a|x, R)} - \Phi(\theta) + \Phi(\theta^*). \tag{43}$$

Also note that $M_\theta$ is locally bounded in the sense that, for every $\ell^2$-ball $U$ in $\Theta$ with a sufficiently small radius $\rho > 0$,

$$\mathbb{E}_{t \sim \text{Unif}[H]}^\beta \left[\sup_{\theta \in U} M_\theta(a_t|x_t, \hat{\boldsymbol{r}})\right] \tag{44}$$

$$\le \mathbb{E}_{t \sim \text{Unif}[H]}^\beta \left[M_{\theta_0}(a_t|x_t, \hat{\boldsymbol{r}}k) + \rho \sup_{\theta \in U} \|\psi_\theta(a_t|x_t, \hat{\boldsymbol{r}})\|_2\right] \tag{45}$$

$$\le \rho\sqrt{\mathbb{E}_{t \sim \text{Unif}[H]}^\beta \left[\sup_{\theta \in U} \|\nabla_\theta M_\theta(a_t|x_t, \hat{\boldsymbol{r}})\|_2^2\right]} < \infty, \tag{46}$$

where $\theta_0$ is the center of $U$. Here, the first inequality follows from $M_\theta(\cdot|\cdot) = M_{\theta_0}(\cdot|\cdot) + \int_0^1 (\theta - \theta_0)^\top \psi_{(1-t)\theta_0 + t\theta}(\cdot|\cdot)\mathrm{d}t$, while the second inequality follows from that $\mathbb{E}^\beta_{t\sim\mathrm{Unif}[H]}[M_\theta(a_t|x_t, \hat{r})] \leq 0$ and Jensen's inequality. This, with Assumption 15 i,ii), allows us to use Theorem 5.14 in [52] and obtain the consistency of MLE: $\hat{\theta} \xrightarrow{P} \theta^*$. Furthermore, with Assumption 15 iii-v), it is possible to use Theorem 5.23 in [52] and have the asymptotic normality

$$\sqrt{n}\left(\hat{\theta} - \theta^*\right) \rightsquigarrow \mathcal{N}(0, \mathcal{I}_{\theta^*}^{-1}\mathcal{V}_{\theta^*}\mathcal{I}_{\theta^*}^{-1}). \tag{47}$$

Finally, we apply the functional delta method (Theorem 20.8 in [52]) on $\hat{\theta}$ and the mapping $\theta \mapsto \{J_j(\pi_{\theta,R})\}_{j,R}$. The desired result follows from calculating the derivative

$$\nabla_\theta J_j(\pi_{\theta,R}) = \sum_{t=1}^H \mathbb{E}^{\pi_{\theta,R}}\left[Q_j^{\pi_{\theta,R}}(x_t, a_t)\nabla_\theta \ln p_\theta(a_t|x_t, R)\right] = H^2 U_{\theta,j}(R), \tag{48}$$

according to the policy gradient theorem (Corollary 3).

# I Lemmas

**Lemma 3.** *Suppose* (25) *holds. Then, we have Assumption 12 with* $C_R \leq K + 1$.

*Proof.* Let $N := B_\infty(R, 1) \cap [0, H]^m$ and note that $\rho := \sup_{R' \in N} \|R' - R\|_\infty \geq 1$. Then, by the assumption, we have

$$1 \geq \int_N f(R'|x)\mathrm{d}R' \geq \rho\{f(R|x) - K\} \geq f(R|x) - K. \tag{49}$$

Rearranging the terms, we get the desired result. $\qquad\square$

**Lemma 4.** *Let* $\epsilon := \epsilon_r + \epsilon_s$. *Then, under Assumption 9, we have*

$$\left\|\hat{P}_T(s, a) - P_T(s, a)\right\|_{\mathrm{TV}} \leq \epsilon \tag{50}$$

*for all* $s \in \mathcal{S}$ *and* $a \in \mathcal{A}$.

*Proof.* It is shown by

$$
\begin{aligned}
\left\|\hat{P}_T(s, a) - P_T(s, a)\right\|_{\mathrm{TV}} &= \sup_E \left|\int_E \mathrm{d}\left\{\hat{P}_T - P_T\right\}(r, s'|s, a)\right| \\
&\overset{(a)}{=} 1 - P_T\left\{(r, s') \in \hat{\mathcal{T}}(s, a)|s, a\right\} \\
&\overset{(b)}{\leq} P_T\left\{\|r - \hat{r}(s, a)\|_\infty > \delta \,|\, s, a\right\} + P_T\left\{s' \neq \hat{s}'(s, a)\,|\,s, a\right\} \\
&\overset{(c)}{\leq} \epsilon,
\end{aligned}
$$

where (a) follows from taking $E = \hat{\mathcal{T}}(s, a)$, (b) from the union bound, and (c) from Assumption 9. $\qquad\square$

**Lemma 5.** *Suppose Assumptions 10 and 13 hold. Then, for all* $x_{t+1} \in \mathcal{X}$ *such that* $(r_t, s_{t+1}) \in \hat{\mathcal{T}}(s_t, a_t)$, *we have*

$$\hat{f}(R|x_t, a_t) - f(R|x_{t+1}) \leq c_R(\delta). \tag{51}$$

*Proof.* Recall that $\hat{f}(\mathbf{R}|x_t, a_t) := \int f(\mathbf{R}|x'_{t+1}) \mathrm{d}\hat{P}_T(\mathbf{r}'_t, s'_{t+1}|x_t, a_t)$, where $x'_{t+1} = (x_t, a_t, \mathbf{r}'_t, s'_{t+1})$. Now, the claim is shown by

$$\hat{f}(\mathbf{R}|x_t, a_t) - f(\mathbf{R}|x_{t+1})$$

$$= \int \left\{ f(\mathbf{R}|x'_{t+1}) - f(\mathbf{R}|x_{t+1}) \right\} \mathrm{d}\hat{P}_T(\mathbf{r}'_t, s'_{t+1}|x_t, a_t)$$

$$\stackrel{(a)}{=} \int \left\{ f(\mathbf{R} - \mathbf{r}'_t + \mathbf{r}_t|x_{t+1}) - f(\mathbf{R}|x_{t+1}) \right\} \mathrm{d}\hat{P}_T(\mathbf{r}'_t, s'_{t+1}|x_t, a_t)$$

$$\stackrel{(b)}{\leq} \sup_{\|\mathbf{r}'_t - \mathbf{r}_t\|_\infty \leq 2\delta} \left\{ f(\mathbf{R} - \mathbf{r}'_t + \mathbf{r}_t|x_{t+1}) - f(\mathbf{R}|x_{t+1}) \right\}$$

$$\stackrel{(c)}{\leq} c_{\mathbf{R}}(\delta),$$

where (a) follows from Assumption 10 and $s'_{t+1} = \hat{s}'(s_t, a_t) = s_{t+1}$ almost surely, (b) from $\|\mathbf{r}_t - \hat{\mathbf{r}}(s_t, a_t)\|_\infty \leq \delta$ and $\|\mathbf{r}'_t - \hat{\mathbf{r}}(s_t, a_t)\|_\infty \leq \delta$ almost surely, and (c) from Assumption 13. $\square$

**Lemma 6.** *We have*

$$\mathbf{J}(\pi) - \mathbf{J}(\pi') = \sum_{t=1}^{H} \mathbb{E}^{\pi'} \left[ \mathbf{Q}^\pi(x_t, \pi(x_t)) - \mathbf{Q}^\pi(x_t, \pi'(x_t)) \right], \tag{52}$$

*where $\mathbf{Q}^\pi(x, a) := \mathbb{E}^\pi[\sum_{h=t}^{H} \mathbf{r}_h|x_t = x, a_t = a]$ is the action value function of $\pi$.*

*Proof.* We may write $\mathbf{Q}^\pi(x, \pi'(x)) := \mathbb{E}_{a \sim \pi'(x)} [\mathbf{Q}^\pi(x, a)]$. Now, observe

$$\mathbf{J}(\pi') = \sum_{t=1}^{H} \mathbb{E}^{\pi'} [\mathbf{r}_t] \tag{53}$$

and

$$\mathbf{J}(\pi) = \mathbf{Q}^\pi(x_1, \pi(x_1)) = \mathbb{E}^{\pi'} \left[ \mathbf{Q}^\pi(x_1, \pi(x_1)) \right] \tag{54}$$

$$= \sum_{t=1}^{H} \mathbb{E}^{\pi'} \left[ \mathbf{Q}^\pi(x_t, \pi(x_t)) - \mathbf{Q}^\pi(x_{t+1}, \pi(x_{t+1})) \right], \tag{55}$$

where the last equality is due to $\mathbf{Q}^\pi(x_{H+1}, \cdot) = 0$. Taking the difference, we see

$$\mathbf{J}(\pi) - \mathbf{J}(\pi') = \sum_{t=1}^{H} \mathbb{E}^{\pi'} \left[ \mathbf{Q}^\pi(x_t, \pi(x_t)) - \mathbf{r}_t - \mathbf{Q}^\pi(x_{t+1}, \pi(x_{t+1})) \right] \tag{56}$$

$$= \sum_{t=1}^{H} \mathbb{E}^{\pi'} \left[ \mathbf{Q}^\pi(x_t, \pi(x_t)) - \mathbf{Q}^\pi(x_t, \pi'(x_t)) \right] \tag{57}$$

where the last equality follows from $\mathbf{Q}^\pi(x_t, a_t) = \mathbb{E}^\pi [\mathbf{r}_t + \mathbf{Q}^\pi(x_{t+1}, \pi(x_{t+1}))|x_t, a_t]$. $\square$

**Corollary 3.** *Suppose Assumption 8 holds. Let $\pi_\theta : \mathcal{X} \to \Delta(\mathcal{A})$ be a policy associated with a parametrized density $p_\theta(a|x)$, $\theta \in \Theta \subset \mathbb{R}^d$, whose score function $\dot{\ell}_\theta(a|x) := \nabla_\theta \ln p_\theta(a|x)$ is bounded in the sense $\mathbb{E}^{\pi_\theta}[\sup_{\theta' \in U} \|\dot{\ell}_{\theta'}(a|x)\|_2] < \infty$ for some $U$ being a neighborhood of $\theta$. Then, we have*

$$\nabla_\theta \mathbf{J}(\pi_\theta) = \sum_{t=1}^{H} \mathbb{E}^{\pi_\theta} \left[ \mathbf{Q}^{\pi_\theta}(x_t, a_t) \dot{\ell}_\theta(a_t|x_t) \right]. \tag{58}$$

*Proof.* Let $\omega > 0$ and fix $\lambda \in \mathbb{R}^d$ arbitrarily. Set $\pi = \pi_{\theta + \omega \lambda}$ and $\pi' = \pi_\theta$, and let $\nu$ be the base measure on $\mathcal{A}$ relative to which $p_\theta(a|s)$ is defined. Now, divide both sides of (52) by $\omega$, and take the

limit $\omega \to 0$ to obtain

$$\lambda^\top \nabla_\theta \boldsymbol{J}(\pi_\theta) = \sum_{t=1}^{H} \lim_{\omega \to 0} \mathbb{E}^{\pi_\theta} \left[ \int \boldsymbol{Q}^{\pi_\theta}(x_t, a) \frac{p_{\theta+\omega\lambda}(a|x_t) - p_\theta(a|x_t)}{\omega} \mathrm{d}\nu(a) \right] \tag{59}$$

$$= \sum_{t=1}^{H} \mathbb{E}^{\pi_\theta} \left[ \int \boldsymbol{Q}^{\pi_\theta}(x_t, a) p_\theta(a|x_t) \lambda^\top \dot{\ell}_\theta(a|x_t) \mathrm{d}\nu(a) \right], \tag{60}$$

where the last equality is owing to the interchange of the expectation and the limit enabled by the dominated convergence theorem. Now, the desired result is shown since $\lambda$ is arbitrary. $\qquad \square$

## J  Proofs of Theorems 2 and 3

**Lemma 7.** *Pick* $\Delta \in (0,1)$ *and set* $\alpha_{\diamond,j} = \sqrt{2\log(|\mathcal{Z}| j^2 \Pi^2 / (6\Delta))}$ *for* $\diamond \in \{r, g\}$. *Then,*

$$|J_\diamond(\pi_{\boldsymbol{z}}) - \mu_{\diamond,j}(\boldsymbol{z})| \le \alpha_{\diamond,j} \cdot \sigma_{\diamond,j}(\boldsymbol{z}) \quad \forall \boldsymbol{z} \in \mathcal{Z} \quad \forall j \ge 1 \tag{61}$$

*holds with a probability at least* $1 - \Delta$.

*Proof.* See Lemma 5.1 and its proof in Srinivas et al. [45]. $\qquad \square$

**Lemma 8.** *Pick* $\Delta \in (0,1)$ *and set* $\alpha_{\diamond,j} = \sqrt{2\log(|\mathcal{Z}| j^2 \Pi^2 / (6\Delta))}$ *for* $\diamond \in \{r, g\}$. *Then, the following inequality holds:*

$$\sum_{j=1}^{N} \left( J_\diamond(\pi_{\boldsymbol{z}^\star}) - J_\diamond(\pi_{\boldsymbol{z}_j}) \right)^2 \le \frac{8}{\log(1 + \nu_\diamond^{-2})} \cdot \alpha_{\diamond,N}^2 \xi_{\diamond,N} \tag{62}$$

*with a probability at least* $1 - \Delta$, *where* $N$ *is the number of iterations in the reward maximization phase.*

*Proof.* This lemma directly follows from Lemma 5.4 in Srinivas et al. [45]. $\qquad \square$

### J.1  Proof of Theorem 2

*Proof.* PLS chooses the next target returns $\boldsymbol{z}$ such that

$$u_{g,j}(\boldsymbol{z}) + L \cdot d(\boldsymbol{z}, \boldsymbol{z}') \le b. \tag{63}$$

By Lemma 7 and the Lipschitz continuity, we have

$$u_{g,j}(\boldsymbol{z}) + L \cdot d(\boldsymbol{z}, \boldsymbol{z}') \ge J_g(\pi_{\boldsymbol{z}}) + L \cdot d(\boldsymbol{z}, \boldsymbol{z}') \tag{64}$$

$$\ge J_g(\pi_{\boldsymbol{z}'}). \tag{65}$$

Therefore, we obtained the desired theorem. $\qquad \square$

### J.2  Proof of Theorem 3

*Proof.* We first define an one-step reachability operator with a certain margin $\zeta \in \mathbb{R}_+$ as

$$\widehat{Z}_\zeta(Y) := Y \cup \left\{ \boldsymbol{z} \in \mathcal{Z} \mid \exists \boldsymbol{z}' \in Y, J_g(\boldsymbol{z}') + \zeta + Ld(\boldsymbol{z}', \boldsymbol{z}) \le b \right\}. \tag{66}$$

Then, we can obtain the following reachable set after $N$ iterations:

$$\widehat{Z}_\zeta^N(\mathcal{Z}_0) := \underbrace{\widehat{Z}_\zeta(\widehat{Z}_\zeta \dots (\widehat{Z}_\zeta(\mathcal{Z}_0)) \dots)}_{N \text{ times}}. \tag{67}$$

Here, the optimal target return $\boldsymbol{z}^\star$ in this paper can now be defined as

$$\boldsymbol{z}^\star := \underset{\boldsymbol{z} \in \widehat{Z}_\zeta^\infty(\mathcal{Z}_0)}{\arg\max} J_r(\pi_{\boldsymbol{z}}). \tag{68}$$

Based on Theorem 1 in Sui et al. [49], it is guaranteed that 1) the safe exploration phase in PLS fully expands the predicted safe set (with some margin $\zeta$) and 2) $\zeta$-optimal target return vector $\boldsymbol{z}^\star$ exists

within the safe set, after at most $N_\dagger$ GP samples. Note that $N_\dagger$ is defined as the smallest positive integer satisfying

$$\frac{N_\dagger}{\alpha_{g,N_\dagger}^2 \xi_{g,N_\dagger}} \geq \frac{C_\dagger(|\widehat{Z}_0^\infty(\mathcal{Z}_0)| + 1)}{\zeta^2}, \tag{69}$$

where $C_\dagger \in \mathbb{R}_+$ is a positive constant.

The following proof mostly follows from that of Theorem 2 in Sui et al. [49], but there are differences in how to construct the confidence intervals. Specifically, for the compatibility with Theorem 1, we cannot assume that the functions are endowed with reproducing kernel Hilbert space (RKHS), which leads to a different bound in terms of optimality.

The reward maximization phase in PLS chooses the next sample using the upper confidence bound in terms of reward within the fully expanded safe region. Thus, by the Cauchy-Schwarz inequality, we have

$$\left( \sum_{j=1}^N \left( J_r(\pi_{\boldsymbol{z}^\star}) - J_r(\pi_{\boldsymbol{z}_j}) \right) \right)^2 \leq N \cdot \sum_{j=1}^N \left( J_r(\pi_{\boldsymbol{z}^\star}) - J_r(\pi_{\boldsymbol{z}_j}) \right)^2 \tag{70}$$

By combining the above inequality with Lemma 8, we have

$$\left( \sum_{j=1}^N \left( J_r(\pi_{\boldsymbol{z}^\star}) - J_r(\pi_{\boldsymbol{z}_j}) \right) \right)^2 \leq N \cdot \frac{8}{\log(1 + \nu_\diamond^{-2})} \cdot \alpha_{r,N}^2 \xi_{r,N} \tag{71}$$

$$= \frac{16 N \xi_{r,N}}{\log(1 + \nu_r^{-2})} \log \left( \frac{|\mathcal{Z}| \Pi^2 N^2}{6\Delta} \right). \tag{72}$$

Given $N_\sharp$ be the smallest positive integer $N$ such that

$$4 \sqrt{\frac{\xi_{r,N}}{N \log(1 + \nu_r^{-2})} \log \left( \frac{|\mathcal{Z}| \Pi^2 N^2}{6\Delta} \right)} \leq \mathcal{E}, \tag{73}$$

we then have

$$\frac{1}{N_\sharp} \sum_{j=1}^{N_\sharp} \left( J_r(\pi_{\boldsymbol{z}^\star}) - J_r(\pi_{\boldsymbol{z}_j}) \right) \leq \mathcal{E}. \tag{74}$$

The LHS of (74) represents the average regret. Thus, there exists $\hat{z} \in \mathcal{Z}$ in the samples such that $J_r(\pi_{\hat{z}}) \geq J_r(\pi_{\boldsymbol{z}^\star}) - \mathcal{E}$. $\qquad\square$

## K  Experiment Details and Additional Results

### K.1  Computational Resources

Our experiments were conducted in a workstation with Intel(R) Xeon(R) Silver 4316 CPUs@2.30GHz and 1 NVIDIA A100-SXM4-80GB GPUs.

### K.2  Hyperparameters

We use the OSRL library[6] for implementing most of the baseline algorithm. We leverage the default hyperparameters used in the OSRL library for the baselines. For CCAC, we use the authors' implementation[7]. For baselines, we use Gaussian policies with mean vectors given as the outputs of neural networks, and with variances that are separate learnable parameters. The policy networks and Q networks for all experiments have two hidden layers with ReLU activation functions. The $K_P$, $K_I$ and $K_D$ are the PID parameters [47] that control the Lagrangian multiplier for the Lagrangian-based algorithms (i.e., BCQ-Lag and BEAR-Lag). We use the same $10^5$ gradient steps and rollout length which is the maximum episode length for CDT and baselines for fair comparison. Specifically, we set the rollout length to 500 for Ant-Circle, 200 for Ant-Run, 300 for Car-Circle and Drone-Circle, 200 for Drone-Run, and 1000 for Velocity. The safe cost thresholds for baselines are 20 and 40 across all the tasks. The hyperparameters used in the experiments are shown in Table 3.

---

[6]https://github.com/liuzuxin/OSRL
[7]https://github.com/BU-DEPEND-Lab/CCAC

Table 3: Hyperparameters for BCQ-Lag, BEAR-Lag, CPQ, COptiDICE, and CCAC.

| Parameter | BCQ-Lag | BEAR-Lag | CPQ | COptiDICE | CCAC |
|---|---|---|---|---|---|
| Actor hidden size | | | [256, 256] | | |
| Critic hidden size | | | [256, 256] | | |
| VAE hidden size | [400, 400] | [400, 400] | [400, 400] | – | [512, 512, 64, 512, 512] |
| $[K_P, K_I, K_D]$ | [0.1, 0.003, 0.001] | [0.1, 0.003, 0.001] | – | – | – |
| Batch size | 512 | 512 | 512 | 512 | 512, 2048 (Velocity) |
| Actor learning rate | 1.0e-3 | 1.0e-3 | 1.0e-4 | 1.0e-4 | 1.0e-4 |
| Critic learning rate | 1.0e-3 | 1.0e-3 | 1.0e-3 | 1.0e-4 | 1.0e-3 |

Moreover, we will present hyperparameters specifically used for the CDT and PLS that are based on return-conditioned supervised learning, in Table 4. The experimental settings are same as the original authors' implementation of CDT.

Table 4: Hyperparameters common for CDT and PLS.

| Parameter | All tasks |
|---|---|
| Number of layers | 3 |
| Number of attention heads | 8 |
| Embedding dimension | 128 |
| Batch size | 2048 |
| Context length $K$ | 10 |
| Learning rate | 0.0001 |
| Droupout | 0.1 |
| Adam betas | (0.9, 0.999) |
| Grad norm clip | 0.25 |

We now summarize the hyperparameters related to GPs in safe exploration and reward maximization phases in PLS. We set the number of episodes for each policy evaluation as $\varpi = 20$ for all tasks. We use GPs with radial basis function (RBF) kernels: one for the reward and one for the safety cost. We set the lengthscales of the reward as 50 for Bullet-Safety-Gym tasks and 100 for Safety-Gymnasium Velocity tasks. The length-scales for the safety cost is set to be 5.0 for all tasks. While variances for the reward are 1.0 for Bullet-Safety-Gym tasks and 100 for Safety-Gymnasium Velocity tasks, those for the safety cost are 1.0 for all tasks. Finally, following Turchetta et al. [51] or Sui et al. [49], we set the Lipschitz constant $L = 0$.

Other important experimental settings include how to set a initial safe set $\mathcal{Z}_0$ associated with Assumption 7. Tables 5 and 6 summarize our experimental settings regarding the initial safe set of target returns.

Table 5: Safe target return range ($\mathcal{Z}_0$) for PLS (Bullet-Safety-Gym).

| Parameter | Ant-Circle | Ant-Run | Car-Circle | Drone-Circle | Drone-Run |
|---|---|---|---|---|---|
| Reward | [250, 300] | [700, 750] | [400, 475] | [700, 720] | [400, 450] |
| Safety | [0, 5] | [0, 5] | [0, 5] | [0, 5] | [0, 5] |

## K.3  Additional Experimental Results

We present additional experimental results for a different threshold $b = 40$ in Table 7. Note that, as for PLS and CDT, the return-conditioned policy in Table 7 is same as that in Table 1. The only

Table 6: Safe target return range ($\mathcal{Z}_0$) for PLS (Safety-Gymnasimum Velocity).

| Parameter | Ant | HalfCheetah | Hopper | Walker2d |
|---|---|---|---|---|
| Reward | [2000, 2300] | [200, 2300] | [1200, 1500] | [2000, 2400] |
| Safety | [0, 5] | [0, 5] | [0, 5] | [0, 5] |

Table 7: Evaluation results for the case with the safety cost threshold $40$. We computed the mean and standard deviation by running each algorithm five times. Reward and cost are normalized; thus, the normalized cost limit is $1.0$. **Bold**: Safe agents whose normalized cost is smaller than 1. Red: Unsafe agents. Blue: Safe agent with the highest reward.

| Task | Metric | BCQ-Lag | BEAR-Lag | CPQ | COptiDICE | CDT | CCAC | PLS |
|---|---|---|---|---|---|---|---|---|
| Ant-Run | Reward ↑ | $0.76 \pm 0.14$ | $0.02 \pm 0.02$ | $0.02 \pm 0.01$ | $0.63 \pm 0.05$ | $0.72 \pm 0.03$ | $0.02 \pm 0.01$ | $0.70 \pm 0.02$ |
| | Safety cost ↓ | $2.34 \pm 0.61$ | $0.05 \pm 0.03$ | $0.00 \pm 0.00$ | $0.56 \pm 0.34$ | $1.10 \pm 0.00$ | $0.00 \pm 0.00$ | $0.54 \pm 0.09$ |
| Ant-Circle | Reward ↑ | $0.78 \pm 0.16$ | $0.63 \pm 0.25$ | $0.00 \pm 0.00$ | $0.17 \pm 0.14$ | $0.53 \pm 0.00$ | $0.62 \pm 0.14$ | $0.55 \pm 0.00$ |
| | Safety cost ↓ | $2.54 \pm 0.87$ | $2.15 \pm 1.38$ | $0.00 \pm 0.00$ | $2.50 \pm 2.81$ | $0.79 \pm 0.00$ | $1.13 \pm 0.44$ | $0.82 \pm 0.00$ |
| Car-Circle | Reward ↑ | $0.79 \pm 0.10$ | $0.84 \pm 0.09$ | $0.73 \pm 0.03$ | $0.49 \pm 0.04$ | $0.80 \pm 0.00$ | $0.77 \pm 0.02$ | $0.80 \pm 0.02$ |
| | Safety cost ↓ | $1.58 \pm 0.38$ | $1.75 \pm 0.37$ | $0.86 \pm 0.04$ | $1.44 \pm 0.72$ | $0.99 \pm 0.05$ | $0.86 \pm 0.04$ | $0.93 \pm 0.06$ |
| Drone-Run | Reward ↑ | $0.68 \pm 0.12$ | $0.87 \pm 0.09$ | $0.19 \pm 0.10$ | $0.69 \pm 0.02$ | $0.60 \pm 0.03$ | $0.57 \pm 0.00$ | $0.62 \pm 0.04$ |
| | Safety cost ↓ | $2.34 \pm 0.64$ | $3.04 \pm 0.61$ | $2.41 \pm 0.34$ | $1.64 \pm 0.10$ | $0.89 \pm 0.11$ | $1.73 \pm 0.01$ | $0.91 \pm 0.09$ |
| Drone-Circle | Reward ↑ | $0.92 \pm 0.05$ | $0.78 \pm 0.06$ | $-0.27 \pm 0.01$ | $0.28 \pm 0.03$ | $0.69 \pm 0.00$ | $0.16 \pm 0.27$ | $0.68 \pm 0.01$ |
| | Safety cost ↓ | $2.31 \pm 0.24$ | $1.69 \pm 0.31$ | $0.20 \pm 0.67$ | $0.29 \pm 0.24$ | $1.00 \pm 0.00$ | $0.71 \pm 0.49$ | $0.96 \pm 0.03$ |
| Ant-Velocity | Reward ↑ | $1.01 \pm 0.01$ | $-1.01 \pm 0.00$ | $-1.01 \pm 0.00$ | $1.00 \pm 0.01$ | $0.97 \pm 0.01$ | $0.60 \pm 0.39$ | $0.99 \pm 0.00$ |
| | Safety cost ↓ | $2.25 \pm 0.29$ | $0.00 \pm 0.00$ | $0.00 \pm 0.00$ | $3.35 \pm 0.74$ | $0.81 \pm 0.44$ | $0.68 \pm 0.29$ | $0.49 \pm 0.05$ |
| Walker2d -Velocity | Reward ↑ | $0.78 \pm 0.00$ | $0.91 \pm 0.03$ | $-0.01 \pm 0.00$ | $0.13 \pm 0.01$ | $0.79 \pm 0.00$ | $0.84 \pm 0.02$ | $0.83 \pm 0.00$ |
| | Safety cost ↓ | $0.30 \pm 0.13$ | $4.05 \pm 1.31$ | $0.00 \pm 0.00$ | $0.90 \pm 0.10$ | $0.00 \pm 0.00$ | $3.49 \pm 0.43$ | $0.00 \pm 0.00$ |
| HalfCheetah -Velocity | Reward ↑ | $1.04 \pm 0.02$ | $0.98 \pm 0.04$ | $0.01 \pm 0.22$ | $0.63 \pm 0.01$ | $0.97 \pm 0.03$ | $0.85 \pm 0.01$ | $1.00 \pm 0.01$ |
| | Safety cost ↓ | $14.10 \pm 3.46$ | $6.34 \pm 5.46$ | $0.10 \pm 0.11$ | $0.00 \pm 0.00$ | $0.05 \pm 0.11$ | $1.22 \pm 0.09$ | $0.01 \pm 0.00$ |
| Hopper -Velocity | Reward ↑ | $0.85 \pm 0.19$ | $0.40 \pm 0.21$ | $0.23 \pm 0.00$ | $0.05 \pm 0.07$ | $0.67 \pm 0.03$ | $0.60 \pm 0.17$ | $0.84 \pm 0.00$ |
| | Safety cost ↓ | $5.30 \pm 3.85$ | $6.08 \pm 3.09$ | $2.75 \pm 0.04$ | $0.46 \pm 0.17$ | $0.56 \pm 0.56$ | $0.60 \pm 0.63$ | $0.20 \pm 0.03$ |

difference regarding PLS between Tables 1 and 7 is the target returns as a result of our target returns optimization algorithm.

Observe that the experimental results in Table 7 exhibit similar tendency to those in Table 1. More specifically, in both cases of $b = 20$ and $b = 40$, PLS is the only method that satisfies the safety constraint in all tasks, while every baseline algorithm violates the safety constraint in at least one task. Moreover, PLS obtains the highest reward return in most tasks, which demonstrates its higher performance in terms of reward and safety.

In addition, we provide Figure 3 to show how our PLS explores target returns $z$. Please observe that PLS guarantees safety in most of policy deployment. Moreover, even if safety constraint is violated, PLS quickly recovers to meet the safety requirement.

### K.4 Online Sample Efficiency

A key advantage of PLS is its sample efficiency during the online optimization phase. Unlike methods that require fine-tuning a high-dimensional policy network, PLS only optimizes a two-dimensional target return vector $(R, G)$. This significantly reduces the number of required online interactions. Our experiments show that PLS typically converges within at most 20 GP iterations. With 20 rollout episodes per iteration for evaluation, this amounts to a total of approximately $400$ online episodes, a number substantially lower than what is typically required for standard policy fine-tuning.

To demonstrate this benefit, we compare PLS against two standard offline-to-online fine-tuning baselines: CDT-FT (S), which uses a small budget of $400$ episodes, and CDT-FT (L), which uses a large budget of $40,000$ episodes. As shown in Table 8, while standard fine-tuning can eventually achieve comparable rewards, it incurs a substantial number of safety violations during the learning process. In contrast, PLS achieves strong performance while maintaining safety throughout, highlighting its suitability for safety-critical applications where online interactions are costly and risky.

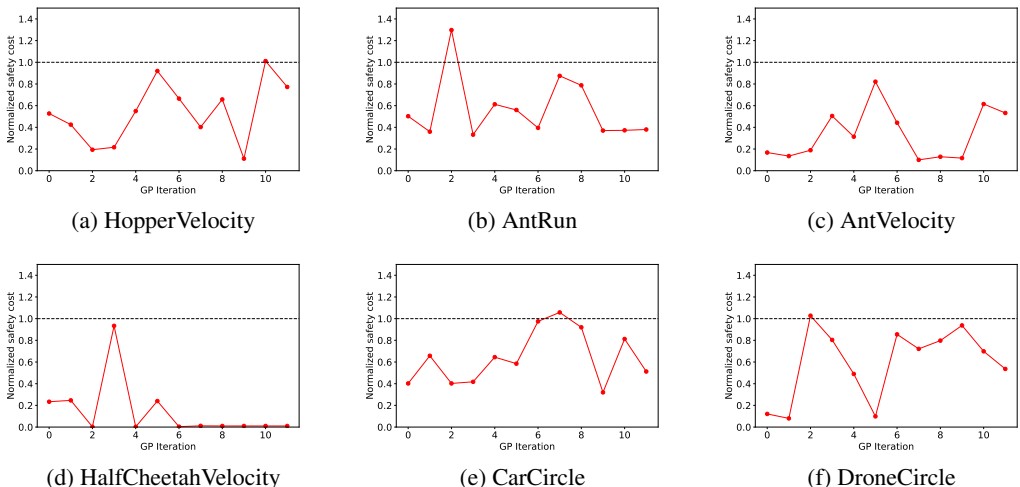

Figure 3: Experimental results on how our PLS ensures the satisfaction of the safety constraint while obtaining new GP observations. Black dotted lines represent the normalized safety threshold.

Table 8: Comparison of PLS and fine-tuning (FT) baselines. PLS achieves comparable final performance to CDT-FT but with significantly fewer safety violations during online adaptation.

| Task | Method | Final Reward ↑ | Final Safety Cost ↓ | Safety Violations during Training ↓ |
|---|---|---|---|---|
| Ant-Run | PLS | $0.78 \pm 0.06$ | $0.77 \pm 0.10$ | $3 \pm 2$ |
| | CDT-FT (S) | $0.75 \pm 0.08$ | $0.80 \pm 0.12$ | $125 \pm 20$ |
| | CDT-FT (L) | $0.80 \pm 0.02$ | $0.90 \pm 0.12$ | $5368 \pm 490$ |
| Ant-Circle | PLS | $0.41 \pm 0.01$ | $0.77 \pm 0.05$ | $2 \pm 1$ |
| | CDT-FT (S) | $0.40 \pm 0.02$ | $0.81 \pm 0.06$ | $98 \pm 15$ |
| | CDT-FT (L) | $0.47 \pm 0.00$ | $1.23 \pm 0.00$ | $10051 \pm 1290$ |
| Car-Circle | PLS | $0.72 \pm 0.01$ | $0.88 \pm 0.09$ | $4 \pm 2$ |
| | CDT-FT (S) | $0.71 \pm 0.03$ | $0.90 \pm 0.11$ | $110 \pm 18$ |
| | CDT-FT (L) | $0.73 \pm 0.01$ | $0.98 \pm 0.12$ | $16023 \pm 2309$ |
| Drone-Run | PLS | $0.59 \pm 0.00$ | $0.50 \pm 0.44$ | $5 \pm 3$ |
| | CDT-FT (S) | $0.58 \pm 0.02$ | $0.55 \pm 0.40$ | $145 \pm 25$ |
| | CDT-FT (L) | $0.59 \pm 0.00$ | $0.82 \pm 0.05$ | $2400 \pm 479$ |
| Drone-Circle | PLS | $0.59 \pm 0.00$ | $0.90 \pm 0.08$ | $3 \pm 2$ |
| | CDT-FT (S) | $0.59 \pm 0.01$ | $0.92 \pm 0.09$ | $85 \pm 14$ |
| | CDT-FT (L) | $0.60 \pm 0.00$ | $0.37 \pm 0.14$ | $3080 \pm 2746$ |
| Ant-Vel | PLS | $0.98 \pm 0.00$ | $0.82 \pm 0.19$ | $2 \pm 1$ |
| | CDT-FT (S) | $0.97 \pm 0.02$ | $0.85 \pm 0.21$ | $130 \pm 22$ |
| | CDT-FT (L) | $0.68 \pm 0.34$ | $0.97 \pm 0.00$ | $17010 \pm 3589$ |
| Walker2d-Vel | PLS | $0.79 \pm 0.00$ | $0.00 \pm 0.00$ | $1 \pm 1$ |
| | CDT-FT (S) | $0.75 \pm 0.04$ | $0.01 \pm 0.01$ | $95 \pm 19$ |
| | CDT-FT (L) | $0.80 \pm 0.00$ | $0.81 \pm 0.07$ | $9810 \pm 2830$ |
| HalfCheetah-Vel | PLS | $0.99 \pm 0.00$ | $0.15 \pm 0.19$ | $1 \pm 1$ |
| | CDT-FT (S) | $0.98 \pm 0.02$ | $0.18 \pm 0.20$ | $160 \pm 30$ |
| | CDT-FT (L) | $0.96 \pm 0.03$ | $0.03 \pm 0.13$ | $2801 \pm 1828$ |
| Hopper-Vel | PLS | $0.83 \pm 0.01$ | $0.42 \pm 0.10$ | $2 \pm 2$ |
| | CDT-FT (S) | $0.82 \pm 0.03$ | $0.45 \pm 0.12$ | $115 \pm 24$ |
| | CDT-FT (L) | $0.84 \pm 0.06$ | $0.82 \pm 0.26$ | $12790 \pm 2589$ |

