# OpenReview forum: "A Provable Approach for End-to-End Safe Reinforcement Learning"
_NeurIPS.cc/2025/Conference — NeurIPS 2025 poster_

### Official Review · Reviewer_A6AP · 2025-06-17

**Clarity:** 3
**Significance:** 3
**Originality:** 3
**Rating:** 5
**Confidence:** 4

**Summary:**

The paper studies an important problem of learning safe policies offline, continuing to adapt online while maintaining safety upon deployment. The paper frames safety with constrained Markov decision processes and proposes using Constrained Decision Transformers (CDT). CDTs are used to train policies that are conditioned on the reward and cost returns, allowing policies to maintain different levels of constraint satisfaction “zero-shot”. The proposed method then uses safe Bayesian optimization techniques to gradually increase the budget, expanding the set of safe policies while maintaining safety.

The paper gives formal safety guarantees, as well as optimality guarantees w.r.t. an optimal policy that can be extracted from the offline dataset. The proposed method (“PLS”) is then evaluated against existing baselines in safe offline RL literature, showing significant improvement in terms safety and performance.

**Questions:**

* How many seeds were used in the experiments? Could you add error bars to figure 3?

**Ethical Concerns:**

["NO or VERY MINOR ethics concerns only"]

**Limitations:**

As discussed above, I believe that the main limitation of the paper is that it tackles a problem in which the offline dataset has high-coverage of the MDP. While the paper makes a solid contribution on this problem, I believe a more crucial (and realistic) challenge for offline-to-online algorithms, especially when safety is a concern, is when the offline dataset is limited, thus online adaptation _of the policy_ is required.

**Quality:**

4

**Strengths And Weaknesses:**

## Strengths
* The connection between (budget-conditioned) CDTs to safe Bayesian optimization is novel and interesting.
* Identifying the bias error between the target reward/budget to the those measured online (Figure 2) is interesting, using a GP to regress the error is correct, and could transfer to similar problems like safe sim-to-real transfer.
* The paper is presented very well.
* Experiments are comprehensive, showing improvement over existing methods.
* The theory seems solid and novel in some aspects.

## Weaknesses

* The proposed method does not in fact continue training the policy with online data. Offline data is generally limited, and online data, being expensive, would not be discarded. Therefore, a more reasonable problem setting would consider the case where the policy adapts/learns while data is collected (similar to [3]). The paper addresses the problem of scalable online end-to-end safety _only_ when the offline dataset has high-enough coverage, meaning that one can extract a good enough policy from it, without ever improving it online. Unfortunately this is generally not the case in practice, with many algorithms that try to address precisely this challenge [1, 2]. I believe this point should be made more clear in the presentation of the paper. While this is mentioned to some extent in the conclusions section, making this limitation more explicit would make the contribution more transparent and will improve the paper.

* The assumption on good data coverage (Assumption 2) might be unrealistic. While this is a fair assumption to make when one theoretically studies the problem, in the context of safety, lack of data coverage, or “distribution mismatch issues between the offline data and the actual environment”, can be detrimental. The paper does not touch upon this limitation.

* In practice, cost and reward return measurements can be quite noisy on real hardware. Assumption 1 covers this case theoretically, however this could be quite limiting, requiring a significant amount of online measurements to fit the GP.

* minor: the method relies on CDTs, making it less practical in problems that require other learning techniques that leverage offline data (e.g. the recent advancements in imitation learning and diffusion/flow-matching models).

[1] Ball, Philip J., Laura Smith, Ilya Kostrikov, and Sergey Levine. "Efficient online reinforcement learning with offline data." In International Conference on Machine Learning, pp. 1577-1594. PMLR, 2023.

[2] Xiao, Wei, Jiacheng Liu, Zifeng Zhuang, Runze Suo, Shangke Lyu, and Donglin Wang. "Efficient Online RL Fine Tuning with Offline Pre-trained Policy Only." arXiv preprint arXiv:2505.16856 (2025).

[3] As, Yarden, Bhavya Sukhija, Lenart Treven, Carmelo Sferrazza, Stelian Coros, and Andreas Krause. "ActSafe: Active Exploration with Safety Constraints for Reinforcement Learning." arXiv preprint arXiv:2410.09486 (2024).

---

> ### Author Rebuttal · Authors · 2025-07-29
>
> We deeply appreciate the reviewer’s encouraging comments. The thoughtful comments of Reviewer A6AP are valuable and help us improve the manuscript's quality. We will answer the Questions and then reply to the comments in Weaknesses.
>
> ### [Question] How many seeds were used in the experiments? Could you add error bars to figure 3?
>
> We used five random seeds for each algorithm and environment in our experiments. Figure 3 in the current manuscript presents a representative result from a single seed to illustrate the safe exploration behavior over GP iterations. In the camera-ready version, we will also add a new figure showing averaged performance (reward and safety cost) with standard deviations across multiple runs, as suggested by the reviewer.
>
> ### [Weakness 1] Offline-to-Online RL
>
> We thank the reviewer for highlighting this important point. We agree that our current method does not update the policy network with online data. This was a deliberate design choice to achieve the primary objective of this work: to propose an end-to-end safe RL algorithm with provable theoretical guarantees.
>
> Our main goal is to ensure the safety of a policy throughout the entire process, from learning to long-term operation. The current problem formulation, which first learns a return-conditioned policy offline and then cautiously optimizes low-dimensional target returns online, is crucial for this goal. This approach allows us to provide high-probability safety guarantees (Theorem 2) and near-optimality guarantees for the target returns (Theorem 3), addressing a longstanding challenge in safe RL.
>
> However, we agree that a natural and valuable extension would be to incorporate online policy updates. Our current framework can serve as a foundation for such an offline-to-online approach. It would be an important future research direction to investigate how to perform continual learning with new online data while still preserving the theoretical safety guarantees. As suggested, we will make this limitation and the potential for future extensions more explicit in the final version of the paper to improve its transparency.
>
>
> ### [Weakness 2] Assumption on good data coverage
>
> Thank you for raising this important point regarding the practicality of Assumption 2. While Assumption 2 is necessary to provide the initial theoretical grounding for the policy trained offline, we agree that assuming good data coverage may be a strong requirement and that a mismatch between the offline data and the online environment is a central challenge in offline RL.
>
> From the perspective of safety guarantee, we would like to clarify that Assumption 2 is only required at the points of interest: the initial target return $z_0$ and the subsequent points $z_N$ selected during early stages of exploration. In practice, by choosing $z_0$ from a region well-covered by the offline dataset, and due to the uncertainty-based selection in PLS, early exploration remains local. This limits reliance on the assumption to neighborhoods where the offline data is dense.
>
> More crucially, our PLS is specifically designed to mitigate the bad effects of this exact "distribution mismatch" during deployment. Specifically, in safe online optimization phase, the role of GPs is precisely to learn the discrepancy between the target returns and the returns achieved online, effectively correcting for the distribution mismatch.
>
> Our PLS uses the GP models to cautiously and safely optimize the policy's behavior in the online environment, with high-probability safety guarantees (Theorem 2). Therefore, rather than ignoring this limitation, our work is a direct attempt to bridge the gap from a potentially limited offline dataset to safe and effective online operation.
>
>
> ### [Weakness 3] Noisy measurements on real hardware
>
> We agree that reward and safety cost return measurements on real hardware can be noisy, and this can affect the number of online samples required to fit an accurate GP model. While our theoretical framework covers this case through Assumption 1 and the noise-aware nature of GPs, we designed our PLS method specifically to be sample-efficient during the online phase. The key to this efficiency lies in the reduced dimensionality of the optimization problem. Unlike conventional RL methods that optimize a high-dimensional policy network, PLS only optimizes a two-dimensional target return vector $z = (R,G)$. As stated on lines 297-299 in the paper, because we are fitting a GP to this simple two-dimensional input space, PLS requires much fewer online interactions than conventional online safe RL algorithms. This low-dimensional optimization significantly reduces the number of online measurements needed to learn the relation between target and actual returns, even in the presence of noise.
>
> ### [Weakness 4, minor] Dependence on CDTs
>
> Thank you for the helpful comment. We agree that our method currently builds on CDTs, which may limit direct applicability in settings favoring alternative offline learning approaches such as imitation learning or diffusion/flow-matching models. However, our framework is grounded in the more general RCSL paradigm, and CDT was chosen as a representative and well-established instance. The structure of CDT enables us to develop a theoretically justified, end-to-end safe RL method. Such theoretical guarantees are currently challenging to establish for alternative paradigms, which typically lack this level of analytical tractability.

---

> > ### Comment · Reviewer_A6AP · 2025-08-02
> >
> > Thank you for your response
> >
> > Some followup questions/suggestions.
> >
> > > [Question] How many seeds were used in the experiments? Could you add error bars to figure 3?
> > We used five random seeds for each algorithm and environment in our experiments. In the camera-ready version, we will also add a new figure showing averaged performance (reward and safety cost) with standard deviations across multiple runs, as suggested by the reviewer.
> >
> > How sensitive is your algorithm to different seeds?
> >
> >
> > > in the paper, because we are fitting a GP to this simple two-dimensional input space, PLS requires much fewer online interactions than conventional online safe RL algorithms. This low-dimensional optimization significantly reduces the number of online measurements needed to learn the relation between target and actual returns, even in the presence of noise.
> >
> > Could you provide more details on the number of online trials needed to learn the GP? Did you calibrate the kernel with offline data? A side-by-side comparison of the number of online trials required to reach near-optimal target returns vs. a standard fine-tuning would reinforce this claim.

---

> ### Author Response · Authors · 2025-08-03
>
> We thank Reviewer A6AP for the thoughtful follow-up and the opportunity to clarify these important practical aspects of our work.
>
> **Sensitivity to random seeds.**
> Our empirical evaluation indicates that PLS is robust to different random seeds. Across five independent runs for each task, we observed consistently low standard deviations in both reward and safety cost, as shown in Tables 1 and 6. This consistency suggests that PLS is not highly dependent on random initialization.
>
> **Number of online trials.**
> For each GP update, we evaluate the current target returns by executing the policy for 20 rollout episodes. PLS typically converges within at most 20 GP iterations. This results in a total of approximately 400 online episodes. This number is significantly lower than what is typically required for fine-tuning an entire policy network, making our approach particularly suitable for safety-critical settings where online interaction is costly. This information was included in Appendix J.2, but given its importance, we will move this discussion to the main experimental section in the camera-ready version to better highlight the sample efficiency of PLS.
>
> **GP kernel calibration.** The GP kernel hyper-parameters (e.g., length-scale and variance) are *not* fixed. Instead, they are learned via marginal likelihood optimization. This approach allows the GP model to flexibly adapt to the observed relationship between target and actual returns in each specific task.
>
> **Side-by-side Comparison.**
> We agree that an explicit, quantitative comparison strongly reinforces our claims. We had, in fact, already conducted such an experiment during the early stages of our research, comparing PLS to two standard offline-to-online fine-tuning (FT) baselines:
>
> - CDT-FT (small, "S"): This baseline uses the same online data budget as PLS (approximately 400 episodes) for fine-tuning.
>
> - CDT-FT (large, "L"): This baseline uses a significantly larger budget of 40,000 online episodes.
>
> The results are summarized in the table below. While both fine-tuning approaches can achieve comparable final performance, they do so at the cost of a **significant number of safety violations during the online learning process.**
>
> | Task | Method | Final Reward $\uparrow$ | Final Safety Cost $\downarrow$ | Safety Violations during Training $\downarrow$ |
> | :--- | :--- | :--- | :--- | :--- |
> | **Ant-Run** | PLS | 0.78 $\pm$ 0.06 | 0.77 $\pm$ 0.10 | 3 $\pm$ 2 |
> | | CDT-FT (S) | 0.75 $\pm$ 0.08 | 0.80 $\pm$ 0.12 | 125 $\pm$ 20 |
> | | CDT-FT (L) | 0.80 $\pm$ 0.02 | 0.90 $\pm$ 0.12 | 5368 $\pm$ 490 |
> | **Ant-Circle** | PLS | 0.41 $\pm$ 0.01 | 0.77 $\pm$ 0.05 | 2 $\pm$ 1 |
> | | CDT-FT (S) | 0.40 $\pm$ 0.02 | 0.81 $\pm$ 0.06 | 98 $\pm$ 15 |
> | | CDT-FT (L) | 0.47 $\pm$ 0.00 | 1.23 $\pm$ 0.00 | 10051 $\pm$ 1290 |
> | **Car-Circle** | PLS | 0.72 $\pm$ 0.01 | 0.88 $\pm$ 0.09 | 4 $\pm$ 2 |
> | | CDT-FT (S) | 0.71 $\pm$ 0.03 | 0.90 $\pm$ 0.11 | 110 $\pm$ 18 |
> | | CDT-FT (L) | 0.73 $\pm$ 0.01 | 0.98 $\pm$ 0.12 | 16023 $\pm$ 2309 |
> | **Drone-Run** | PLS | 0.59 $\pm$ 0.00 | 0.50 $\pm$ 0.44 | 5 $\pm$ 3 |
> | | CDT-FT (S) | 0.58 $\pm$ 0.02 | 0.55 $\pm$ 0.40 | 145 $\pm$ 25 |
> | | CDT-FT (L) | 0.59 $\pm$ 0.00 | 0.82 $\pm$ 0.05 | 2400 $\pm$ 479 |
> | **Drone-Circle**| PLS | 0.59 $\pm$ 0.00 | 0.90 $\pm$ 0.08 | 3 $\pm$ 2 |
> | | CDT-FT (S) | 0.59 $\pm$ 0.01 | 0.92 $\pm$ 0.09 | 85 $\pm$ 14 |
> | | CDT-FT (L) | 0.60 $\pm$ 0.00 | 0.37 $\pm$ 0.14 | 3080 $\pm$ 2746 |
> | **Ant-Vel**| PLS | 0.98 $\pm$ 0.00 | 0.82 $\pm$ 0.19 | 2 $\pm$ 1 |
> | | CDT-FT (S) | 0.97 $\pm$ 0.02 | 0.85 $\pm$ 0.21 | 130 $\pm$ 22 |
> | | CDT-FT (L) | 0.68 $\pm$ 0.34 | 0.97 $\pm$ 0.00 | 17010 $\pm$ 3589 |
> | **Walker2d-Vel**| PLS | 0.79 $\pm$ 0.00 | 0.00 $\pm$ 0.00 | 1 $\pm$ 1 |
> | | CDT-FT (S) | 0.75 $\pm$ 0.04 | 0.01 $\pm$ 0.01 | 95 $\pm$ 19 |
> | | CDT-FT (L) | 0.80 $\pm$ 0.00 | 0.81 $\pm$ 0.07 | 9810 $\pm$ 2830|
> | **HalfCheetah-Vel**| PLS | 0.99 $\pm$ 0.00 | 0.15 $\pm$ 0.19 | 1 $\pm$ 1 |
> | | CDT-FT (S) | 0.98 $\pm$ 0.02 | 0.18 $\pm$ 0.20 | 160 $\pm$ 30 |
> | | CDT-FT (L) | 0.96 $\pm$ 0.03 | 0.03 $\pm$ 0.13 | 2801 $\pm$ 1828 |
> | **Hopper-Vel**| PLS | 0.83 $\pm$ 0.01 | 0.42 $\pm$ 0.10 | 2 $\pm$ 2 |
> | | CDT-FT (S) | 0.82 $\pm$ 0.03 | 0.45 $\pm$ 0.12 | 115 $\pm$ 24 |
> | | CDT-FT (L) | 0.84 $\pm$ 0.06 | 0.82 $\pm$ 0.26 | 12790 $\pm$ 2589 |
>
> We believe these results clearly illustrate the core benefit of our approach. PLS maintains safety throughout online optimization, consistent with our high-probability guarantees. In contrast, standard fine-tuning leads to many unsafe episodes before converging due to its lack of an explicit safe learning mechanism. In the camera-ready version, we will add a new section that includes a side-by-side analysis of the online sample budget required by PLS versus that of a standard offline-to-online fine-tuning approach.
>
> We sincerely appreciate these detailed follow-up questions. They have allowed us to clarify implementation details and better explain the practicality and robustness of PLS.

---

> > ### Comment · Reviewer_A6AP · 2025-08-03
> >
> > Thank you for providing these additional experiment results. I will keep my (already high) score as I believe this work is solid.

---

### Official Review · Reviewer_njCE · 2025-06-26

**Clarity:** 2
**Significance:** 3
**Originality:** 2
**Rating:** 3
**Confidence:** 4

**Summary:**

This paper presents Provably Lifetime Safe RL. The method begins by training a policy / agent in the offline safe RL setting, i.e. learning from a static dataset of state transitions, rewards, and costs under some behavior policy. It then introduces a notion of approximating target returns (again, both rewards and costs) using Gaussian Processes to allow for further optimization via (1) safe exploration, i.e. the finding of an increasingly expansive set of safe trajectories and then (2) reward maximization. The paper provides theoretical analysis and justification for the proposed approach.

**Questions:**

1. Why would alpha_r be equal to alpha_g in (5)? This seems to imply that exploring to increase performance is the same as safety? There is a 2-step scheme for exploration that controls for this somewhat; i.e. a kind of "safe set" is identified first before reward maximization. I am just wondering if, perhaps the allowed failure probabilities can be tuned for safe exploration versus reward maximization (i.e. it is not ok to fail during safe exploration, but it might slightly more ok to fail during reward maximization). In this case, perhaps Delta with a subscript for each phase?

2. GPs have nice properties, many of which this paper takes advantage of. For example, smoothness, continuity, etc, as well as being nonparametric. However, they also have issues. One is that - if one wants to preserve certain theoretical properties - then computational and scaling issues arise as the size of the data grows. The authors seem to have addressed this, for example by assuming k is diagonal, i.e. uncorrelated covariance function. This is somewhat related to the fact that GPs are very sensitive kernel selection and have been found to in fact be poor at uncertainty quantification (which seems to be a motivation of this paper, offline RL in general, and the notion of going from a static dataset to online inference). See Wilson, Andrew, and Ryan Adams. "Gaussian process kernels for pattern discovery and extrapolation." In International conference on machine learning, pp. 1067-1075. PMLR, 2013.For example, deep ensembles may not be amenable to the same theoretical treatment but they simply "work better". This comment digresses somewhat from the main contributions of the paper but the choice of GPs (and perhaps the tricks used to make this work in practice) could be described in more detail.

3. The experimental results appear to be incomplete. This appears to be from Bullet Safety Gym and a few tasks in Safety Gym. Why not every task in Safety Gym and Bullet Safety Gym? And why not Metadrive, a la the following papers (below) and many others?

4. In section 6.1, where do the target return vectors come from? This is often a practical problem with return-conditioned RL, because either the task or reward (cost) functions have to be amendable to knowing a priori what a "good" target. The desired reward per state plus an appropriate episode length might need to be known, for example. The paper states that the next target returns are chosen sequentially that maximize reward return subject to safety constraint. How? To be clear, this question is about the process of collecting (generating) data used for fitting the two GPs. How does one know that this process is safe, for example? NOTE: this question is addressed in section 6.2 but I will leave the comment here because some of the questions are still valid; that is, the problem must still be amenable to the notion of safe exploration in section 6.2 so both the task structure and cost function formulation must be appropriate. Note also a separate question about baselines; perhaps these questions are all related.

5. Is the 'H' in the definition of candidate thresholds (line 116) the same H that defines episode length? It is unclear so far why this would be the case. Also, the notation here is a bit strange. It implies a *set* of candidate thresholds but this does not appear to be a finite set. Is the optimization problem in (1) for an infinite set of constraints?

6. Line 92-93, the episode length is fixed for each episode; does this mean all episodes have the same length? Or what is "fixed" here?

Other comments and questions
The abstract states, "while cautiously optimizing a limited set of parameters, known as target returns". This is a bit confusing. Is part of (for example) the policy network frozen such that only a subset of parameters are updated during deployment? I do not think this is the case; rather, the policy is optimized relative to a limited set of *objectives*.

Line 75-77 states, "offline safe RL still suffers from a central difficulty: learned policies often become either unsafe or overly conservative, largely due to the intrinsic challenges of off-policy evaluation (OPE) in stateful environments". There are certainly challenges in OPE, and they are related to some of the issues in Offline RL, but they are not exactly 1:1. OPE has to do with performance evaluation when one only has access to data. Another way of saying this is that OPE is a kind of *solution* to some of the challenges in offline RL, but an alternative is to train from offline data and then test in in the environment. The latter approach has challenges due to OOD, the dataset used for training being insufficient, etc. In fact, one of the grand challenges is dealing with narrow or biased data distributions; OPE itself can suffer from this. This is not a fundamental issue with the paper but is a bit of a strange representation of the problem.

Is a context-dependent policy Markovian?

This is mostly clear from the context but the paper might be improved by mentioning that everything is in terms of undiscounted returns (or costs).

Many works have gone to great lengths to try to predict returns under a policy - this is essentially one of the pillars of RL. In fact, one small piece of the decision transformer approach is to use it as a critic. GP seems like an interesting approach given that almost every work out there is parametric, and in fact involves some type of deep learning framework.



Yinan Zheng, Jianxiong Li, Dongjie Yu, Yujie Yang, Shengbo Eben Li, Xianyuan Zhan, and Jingjing Liu. Safe offline reinforcement learning with feasibility-guided diffusion model. arXiv preprint arXiv:2401.10700, 2024.
Koirala, Prajwal, Zhanhong Jiang, Soumik Sarkar, and Cody Fleming. "Latent Safety-Constrained Policy Approach for Safe Offline Reinforcement Learning." arXiv preprint arXiv:2412.08794 (2024).

**Ethical Concerns:**

["NO or VERY MINOR ethics concerns only"]

**Final Justification:**

This is a nice paper. Other reviewers rated it higher and appreciated the theoretical contribution perhaps more than I did. In the end, one has to choose a ranking/score, and there is nothing between weak reject and weak accept; if there were, I would probably choose that. As it is, I will leave my score as-is but obviously fully support the other reviewers.

**Limitations:**

While the paper does address limitations (and these limitations are indeed interesting avenues of research), perhaps the questions above and other reviewers' questions will assist the authors in identifying other potentially significant limitations.

**Quality:**

3

**Strengths And Weaknesses:**

Strengths:
- The paper is relevant, i.e. it addresses known gaps or challenges in offline RL, and offline safe RL specifically, in particular the leap from the dataset to the environment and the various issues that arise due to this transition.
- The paper is rigorous theoretically.
- The paper leverages several existing ideas and approaches in clever ways, including pulling together GPs with deep RL.

Weaknesses:
- The assumptions may not match reality, or to what extent do they match reality?
- While interesting, the use of GPs seems to be more theoretically motivated than for performance or computational reasons.
- There is not really an "algorithm", although it is somewhat made clear during the text.
- There is no ablation. The framework starts with CDT, a very strong baseline. Can any approach that first trains a policy and then allows it to explore afterwards -- would this not, by definition, result in performance that is no worse than CDT?

---

> ### Author Rebuttal · Authors · 2025-07-29
>
> We deeply appreciate the reviewer’s thoughtful feedback. The thoughtful comments of Reviewer njCE are valuable and help us improve the manuscript's quality. We will first answer the Questions and address the comments in Weaknesses.
>
> ### [Question 1] Setting of $\alpha$
>
> The primary reason we set $\alpha_r = \alpha_g$ is to clearly convey our key ideas while avoiding complex notations. We agree that using separate failure probabilities is a valuable extension for practical applications and will discuss this in the final version.
>
> ### [Question 2] Challenges of GPs
>
> We thank the reviewer for constructive feedback! While we acknowledge the general challenges with GPs regarding scaling and uncertainty quantification, these concerns are substantially mitigated in our work for two key reasons:
>
> - The primary reason computational and scaling issues are not a concern is that PLS does not operate on the high-dimensional state or state-action spaces. Instead, it optimizes only a two-dimensional target return vector $(R,G)$. The amount of data required for the GP to model this low-dimensional space is very small, making the computational cost negligible.
>
> - Another reason for choosing GPs over alternatives like deep ensembles is that they provide a principled way to quantify uncertainty. This is not just a nice property; it is the fundamental mechanism that allows us to provide the high-probability safety and near-optimality guarantees (Theorems 2 and 3) that are central to our paper's contribution. While a deep ensembles is a nice function approximator, they do not usually offer this same level of provable guarantee for safe exploration.
>
> As the reviewer correctly inferred, our implementation makes a simplifying assumption that the reward and safety cost returns are uncorrelated, allowing us to model them with independent GPs. We agree that the practical details of our GP configuration could be described more thoroughly. In the final version of the paper, we will include a more detailed discussion on our kernel selection and provide a clearer justification.
>
> ### [Question 3] Empirical experiment
>
> Our primary objective in the empirical evaluation was to conduct a rigorous and direct comparison against the current state-of-the-art method for *versatile* offline safe RL, which is Constraint-Conditioned Actor-Critic (CCAC) [Guo et al., 2025]. To conduct the most direct and fair comparison, we have included all the environments used in the original CCAC paper. This allows us to clearly demonstrate the advantages of our proposed method relative to the most relevant and competitive baseline.
>
> - Zijian et al. "Constraint-conditioned actor-critic for offline safe reinforcement learning." ICLR. 2025.
>
> While we acknowledge that other benchmarks like Metadrive exist, our focus was on providing a targeted empirical evaluation that supports our main claims. Crucially, let us highlight that **a core contribution of our work also lies in the theoretical analysis**, where we establish a provable framework for end-to-end safety. The experiments are designed to validate this theoretical foundation and demonstrate its practical effectiveness, rather than to exhaustively cover every possible benchmark. We believe our chosen set of experiments successfully achieves this goal.
>
> ### [Question 4] Target returns in Section 6.2
>
> The target return vectors are selected through a safe exploration process described in Section 6.2. We begin with a conservative initial set (low reward and safety cost targets), which empirically produces safe behavior. These initial rollouts form the seed data for GP modeling.
>
> As proved in Theorem 1, the relationship between target and actual returns can be modeled as a GP with bounded bias. This theoretical result justifies our use of GP-based uncertainty to guide exploration. New target returns are selected only if the GP’s worst-case predicted cost remains below the safety threshold, which ensures all data collection is performed under high-probability safety guarantees (Theorem 2).
>
> ### [Question 5] $H$ in the definition of candidate thresholds
>
> Yes, $H$ is the episode length. We assume the safety threshold $b$ is bounded in $[0,H]$, as the maximum cumulative safety cost is $H$, for avoiding trivial cases. The optimization in (1) finds a single, *versatile* policy, $\pi$, adaptable to a continuous range of safety thresholds, $b$. This contrasts with training a unique policy for each specific threshold. We will make the problem statement more explicit in the revision.
>
> ### [Question 6] Fixed episode length
>
> Yes, in our formal problem statement, "fixed" means that all episodes are defined to have the same constant length, $H$. This is a standard assumption for finite-horizon CMDPs. In practice, we handle variable-length episodes by treating $H$ as the maximum episode length and padding shorter trajectories. This is a standard technique that works seamlessly with return-conditioned sequence models like the CDT.
>
> ### [Weakness 1] The assumptions may not match reality
>
> We agree that theoretical assumptions inevitably introduce some gap between theory and practice. In our case, the assumptions, such as near-deterministic transitions and sufficient data coverage, are adopted for rigorous guarantees while still capturing important characteristics of many real-world safe RL scenarios.
>
> While these conditions may not fully hold in all cases, they are loosely aligned with common assumptions in safe RL literature (e.g., smooth dynamics in simulated robotics). For instance, deterministic state transition is a popular assumption in safe RL literature, as being represented by
>
> - Wachi and Sui. "Safe reinforcement learning in constrained Markov decision processes." ICML, 2020.
>
> - Thomas et al. "Safe reinforcement learning by imagining the near future." NeurIPS. 2021.
>
> That said, we acknowledge that relaxing these assumptions is a valuable direction for future work. We will revise the paper to clarify the practical implications of each assumption.
>
> ### [Weakness 2] While interesting, the use of GPs seems to be more theoretically motivated than for performance or computational reasons.
>
> While our use of GPs is indeed theoretically motivated, we view this as a key strength of our framework, not a limitation or weakness. The ability of GPs to provide calibrated uncertainty estimates is what enables PLS to reason about safety in a principled way. This is essential for enjoying high-probability safety guarantees and for enabling reliable safe exploration under constraints. Our goal is not simply to achieve SOTA performance on benchmarks. Rather, we believe that for the safe RL community, especially with a view towards real-world, safety-critical applications, it is more important to build methods that are reliable and well-understood. **Our primary contribution is a framework where we can explain why and how safety is guaranteed.**
>
> ### [Weakness 3, 4] There is no algorithm/ablation
>
> Thank you for the comments. We would like to highlight that a pseudo-code of PLS has been provided in Appendix B.
> Also, it may seem to be a natural outcome that PLS outperforms the baseline CDT. Our contribution, however, is not simply improving performance, but rather being the first to propose a end-to-end safe RL framework that captures the relationship between target returns and actual returns and applies this to safe online optimization. Crucially, our claims are supported both theoretically and empirically.
>
> We first theoretically demonstrate that the difference between target and actual returns follows a GP (Theorem 1) . This is a non-trivial insight that provides the theoretical foundation for using a GP-based optimization approach. Based on this theoretical insight, we proposed PLS to optimize target returns using GPs. Through experiments, we empirically demonstrated that this framework performs better than baselines.
>
> A simpler search or optimization method could not provide such provable safety guarantees. We believe the value of our work lies in presenting a new approach for safe RL that is validated by both theoretical insight and empirical results.
>
> ### [Other Comments]
>
> **Confusing sentence in Abstract.** Thank you for flagging this ambiguity. We optimize the target return inputs $(R,G)$ to steer a fixed policy, maximizing rewards under safety constraints. We will clarify this in the abstract with your suggested text: "...while cautiously optimizing a limited set of inputs, known as target returns, that condition the policy's behavior."
>
> **OPE and distribution mismatch.**
> Our intention on lines 75-77 was to convey that the difficulty in learning a good policy offline is associated with the difficulty of accurately evaluating the long-term performance from a static dataset. The inability to perform reliable OPE, primarily due to the distributional shift, often leads to overly conservative or unsafe policies. We agree with the reviewer that it would be a more precise representation to frame this issue as a result of distribution mismatch. In the final version of the paper, we will revise the sentence to more accurately reflect this nuance.
>
> **Context-dependent policy.** A context-dependent policy defined in the paper is not necessarily Markovian. We define a context-dependent policy as a more broader class than a state-dependent policy to conduct theoretical analyses of RL policies based on Transformer and the mixture of multiple such policies.
>
> **Undiscounted returns.** Thank you for helpful suggestions. We will add explanations that undiscounted returns or safety costs are considered.
>
> **Interesting approach.** Thank you for this encouraging feedback and for highlighting the novelty of our approach! Our choice of a GP, while non-standard, was directly motivated by our theoretical analysis (Theorem 1). Its ability to provide principled uncertainty bounds is what enables the provable safety guarantees that are central to our work.

---

> > ### Comment · Reviewer_njCE · 2025-08-01
> >
> > First, thank you for the comprehensive rebuttal. Many of my questions and/or sources of confusion were address nicely.
> >
> > I am still stuck on a point that is sort of at the center of the "Venn diagram" of Question 2 (i.e the second part of your response related to UQ properties of GPs), Question 3, Weakness 1, and Weakness 2. Let me try to unpack this.
> >
> > Any theoretical approach has to make assumptions. Sometimes these assumptions hold well, sometimes the assumptions break down. And in both cases, the practical outcomes are not always intuitive; for example, theoretical assumptions hold but the empirical results don't do very well, or the opposite, that theoretical assumptions break down but things still "work" in practice.
> >
> > This is why I was asking for more empirical evaluation. Different people with different perspectives may disagree on what is sufficient (indeed on what is sufficient for theory in addition to what is sufficient empirically). With all of this said, it still _feels_ incomplete to me but I do appreciate the explicit framing of the paper as doing both theoretical and empirical work.

---

> ### Author Response · Authors · 2025-08-02
>
> Dear Reviewer njCE,
>
> Thank you for your thoughtful follow-up comment and for engaging in this constructive discussion. Your feedback has been invaluable in helping us reflect more deeply on the core contributions and limitations of our paper.
>
> We understand that your central remaining concern lies in the "Venn diagram" of our theoretical assumptions and their practical outcomes, and we agree that this is a critical and fundamental point in the field of safe RL. We recognize your perspective that further empirical evaluation would help bridge this gap.
>
> Our primary goal with this paper was to introduce a framework that offers provable safety guarantees, which we believe is a crucial step for real-world, safety-critical applications. We focused our experiments on a direct and rigorous comparison with the SOTA to validate our core claim: that *PLS can achieve end-to-end safety where other methods fail, supported by a solid theoretical foundation*.
>
> In response to your valuable feedback and to address the perceived incompleteness, we commit to making the following improvements in the camera-ready version of the paper:
>
> **Deeper discussion on the role of assumptions.** We will add a more detailed discussion on the relationship between our theoretical assumptions (e.g., near-deterministic transitions) and the empirical settings. We will analyze why PLS performs robustly even when these assumptions may not be perfectly met in practice, addressing your insightful point that theoretical assumptions can sometimes break down while the method still works. This will help clarify the practical robustness of our framework.
>
> **Expanded limitations and future work section.** We will expand the limitations section to more explicitly discuss the gap between our current theoretical guarantees and performance in more complex, highly stochastic environments. We will outline concrete future research directions aimed at relaxing our current assumptions and extending the empirical validation to a wider range of benchmarks, as you suggested.
>
> We believe that pursuing both theoretical rigor and practical utility is essential for advancing the field of safe RL. Your feedback has provided us with a crucial perspective on how to better balance these two aspects in our paper.
>
> Thank you once again for your constructive and insightful comments, which will undoubtedly help us to improve the quality and transparency of our manuscript.
>
> Best regards,
>
> Authors

---

### Official Review · Reviewer_XNH5 · 2025-07-01

**Clarity:** 3
**Significance:** 2
**Originality:** 2
**Rating:** 4
**Confidence:** 3

**Summary:**

This paper studies a mix of lifelong learning with safe reinforcement learning. The paper proposes Provably Lifetime Safe Reinforcement Learning (PLS), with the aim of ensuring safety throughout the entire RL process from learning to deployment. PLS combines offline safe RL using return-conditioned supervised learning (specifically Constrained Decision Transformer) with online safe policy deployment. The main goal is to find the optimal pair of target returns $\boldsymbol{z}=(R, G)$ that maximizes $J_r\left(\pi_{\boldsymbol{z}}\right)$ while guaranteeing the satisfaction of the safety constraint (i.e., $J_g\left(\pi_{\boldsymbol{z}}\right) \leq b$ ) according to GP-based inferences. For this purpose, they optimistically sample the next target returns $\boldsymbol{z}$ while pessimistically ensuring the satisfaction of the safety constraint. The key innovation is optimizing target returns (reward and safety cost) using Gaussian Processes (GPs) during deployment, rather than using fixed target returns. The authors provide theoretical analysis showing the relationship between target and actual returns follows a Gaussian process with a small bias term. They prove that PLS guarantees safety with high probability throughout the process and finds near-optimal target returns. Experimentally, PLS is evaluated on continuous robot locomotion tasks from the Bullet-Safety-Gym and Safety-Gymnasium benchmarks. Results show PLS is the only method that satisfies safety constraints across all tasks while achieving competitive rewards. The method requires fewer online interactions than conventional online safe RL due to optimizing only two-dimensional target returns. The approach assumes access to an offline dataset and initial safe target returns, leveraging GPs' uncertainty estimates for safe exploration.

**Questions:**

I have the following questions:

1. Even though this paper tries to set itself apart from the existing safe RL works, there are some similarities. For example, in assumption 7, line 278 there exists the safe set. This basically aids in the intial exploration which is reflected in Theorem 2. How is this approach different from other existing safe RL works?
2. Where is the pessimistic exploration showing up? It seems to me from section 6.2 lines 263-268 that to conduct safe exploration, similar to Srinivas et al. you are also using optimistic GP UCB?
3. The termination condition also requires more explanation. Safe exploration is terminated under the condition $\max\_{\boldsymbol{z} \in E_N}\left(u\_N(\boldsymbol{z})-\right.$ $\left.\ell_N(\boldsymbol{z})\right) \leq \zeta$, where $\zeta \in \mathbb{R}_{+}$is a tolerance parameter. How is this tolerance parameter chosen? Shouldn't this be a problem-specific parameter that should depend on the hardness of the problem?
4. Why did you choose to use CDT as the base offline RL algorithm? Have you experimented with other return-conditioned methods, and would the GP-based optimization still be effective with different underlying policies?
5. Your experiments show some violations of safety constraints during exploration (Figure 3), despite theoretical high-probability guarantees. Can you elaborate on the sources of this gap and whether tighter bounds or different GP configurations could reduce these violations?

**Ethical Concerns:**

["NO or VERY MINOR ethics concerns only"]

**Limitations:**

See weakness and questions section.

**Paper Formatting Concerns:**

Not applicable.

**Quality:**

3

**Strengths And Weaknesses:**

Strengths

1. The paper provides a rigorous theoretical analysis establishing the relationship between target and actual returns through Theorem 1, showing it follows a Gaussian process with explicit bias terms. The safety guarantees (Theorem 2) and near-optimality results (Theorem 3) are formally proven with clear assumptions (Assumptions 1 to 4, lines 170-186).
2. To me, the combination of offline constrained safe  RL with online GP-based optimization is well-motivated (but the exact novelty is not clear). This tries to address a current limitation of existing safe RL paradigms, where online methods struggle with initial safety and offline methods suffer from distribution mismatch.
3. The experiments cover 9 different tasks across two benchmarks, with consistent results showing PLS as the only method maintaining safety across all tasks while achieving competitive rewards. The computational efficiency advantage is also demonstrated.

Weaknesses

1. The assumptions 1-4 are standard and prima facie seem to stem from other offline RL and safe RL literature. It is not clear what the key additional requirement is needed for this CSRL setting.
2. Theorem 3 only guarantees near-optimal target returns, not a near-optimal policy. This gap between optimizing target returns and achieving optimal policy performance is not addressed.
3. Going back to the assumptions, the theoretical analysis relies on restrictive assumptions, including near-deterministic transitions (Assumption 1), specific conditions on return density functions, and soft realizability. Not only may these not hold in many practical scenarios, but even their own empirical setting doesn't satisfy this.
4. The GP bandit motivation of Srinivas et al. (2014) specifically tackled high high-dimensional setting. However, their approach is limited to single safety constraints (two-dimensional target returns). Extension to multiple constraints or high-dimensional return spaces would face computational challenges and may require fundamentally different approaches.

---

> ### Author Rebuttal · Authors · 2025-07-29
>
> We deeply appreciate the reviewer’s encouraging comments. The thoughtful comments of Reviewer XNH5 are valuable and help us improve the manuscript's quality. We will answer the Questions first and then address the comments in Weakness.
>
> ### [Question 1] Differences from other existing safe RL works
>
> Thank you for the thoughtful question. The key difference lies in how the input space or initial safe set is defined. In most existing methods, the safe set is defined over high-dimensional states or state-action pairs, which makes their construction and verification challenging. In contrast, our method assumes a safe set over target returns. This space is low-dimensional (typically two-dimensional) and allows direct control over the behavior of a return-conditioned policy. By setting the target returns to low values for both reward and safety cost, the policy tends to output generally safe actions, so identifying target returns that belong to the initial safe set is not difficult.
>
> ### [Question 2] Pessimistic exploration
>
> Thank you for the helpful question. Because the safety constraint is represented as $J_g(\pi) \le b$, incorporating the upper bound of the safety cost function is pessimistic. By constructing $Y_N$ using the upper bound of the safety function, we aim to explore the safety cost function within the pessimistically identified safe region. We acknowledge that this may not have been sufficiently clear in the current text and will add further clarification in the camera-ready version.
>
> ### [Question 3] Termination condition
>
> Thank you for the important question. We agree that the choice of the tolerance parameter plays an important role and may be problem-specific. In our experiments, however, we found that using a small fixed value exhibited stable and effective performance across all evaluated tasks. While this approach may not be universally optimal, it provided a practical and consistent stopping criterion in our settings. We will add further explanation and discussion of this choice in the camera-ready version.
>
> ### [Question 4] CDT as the base offline RL algorithm
>
> Thank you for this insightful question. One of our main contributions in this paper lies in the theoretical analysis presented in Section 5, which is specifically developed for the (constrained or multi-objective) Decision Transformer. These theoretical results do not directly extend to arbitrary return-conditioned methods. Therefore, our choice of CDT was primarily motivated by the need to align the theoretical framework with the empirical implementation. That said, as the reviewer suggests, we believe that our PLS can also be empirically effective when combined with other return-conditioned policies, and exploring this generality would be an interesting direction for future work.
>
> ### [Question 5] Some safety violations during exploration
>
> Thank you for the thoughtful question. Crucially, our safety guarantee in Theorem 2 is *probabilistic*. This means that a small number of constraint violations are theoretically permissible and expected in rare cases. The primary source of the gap is the approximation error between the GP model and the (complex and noisy) true safety cost return. Also, the finite number of rollout evaluations introduces noise that can affect the GP's predictions.
>
> As the reviewer suggests, it is possible to reduce the number of violations. For example, using more conservative confidence bounds (e.g., increasing $\alpha_g$) or carefully tuning the GP kernel and its hyper-parameters could lead to more cautious exploration and better alignment with the safety constraints, at the cost of exploration efficiency. We will include a discussion of these points in the camera-ready version.
>
> ### [Weakness 1] Key additional requirement needed for our RCSL setting
>
> Assumption 2 (initial coverage) is conceptually similar to the concentrability condition often used in offline RL to ensure that important target distributions are well-supported by the dataset.
> Assumptions 1, 3, and 4, on the other hand, are more specific to the analysis of CDT in our setting. These assumptions help us characterize the relationship between the target returns and the actual returns, which is critical for ensuring that the GP model used in PLS poses sufficient accuracy. In particular, they are used to show that the CDT policy is approximately unbiased, which supports our theoretical guarantees. We will add more clarifications in Appendix C.2.
>
> ### [Weakness 2] Theorem 3 only guarantees near-optimal target returns, not a near-optimal policy
>
> Thank you for the insightful feedback! As noted in our Limitations paragraph, our PLS does not guarantee the optimality of the resulting policy. This gap stems from the limitations of the underlying RCSL framework, which currently lacks a theoretical characterization of how closely it approximates optimal policies. While PLS benefits from provable safety and efficient target return optimization, it also inherits this fundamental gap from RCSL. Bridging this gap would require a deeper theoretical understanding of how well RCSL approximates optimal policies under given target returns, and we consider that this direction is highly valuable for future work.
>
> ### [Weakness 3] Restrictive assumptions
>
> Thank you for pointing this out. As discussed earlier, some of the assumptions (i.e., Assumptions 1, 3, 4, and 5) are used solely to bound the bias in the RCSL model and ensure that the GP approximation remains sufficiently accurate. Importantly, our theoretical results still hold even when the GP is biased; that is, this is captured by the bias term $\varepsilon(z)$ in Theorem 1, which quantifies how deviations from ideal assumptions affect the predictive accuracy.
>
> In particular, while Assumption 1 (near-deterministic transitions) is  strong, it is more general than the one used in Brandfonbrener et al. [9], where exact deterministic transitions (i.e., $\delta = 0$) are assumed. Our formulation allows for a $\delta$-neighborhood around deterministic dynamics, which permits small stochastic perturbations and thus covers a broader class of environments. Moreover, similar (near-)deterministic assumptions are common in the safe RL literature, as seen in:
>
> - Berkenkamp et al. "Safe model-based reinforcement learning with stability guarantees." NeurIPS, 2017.
>
> - Wachi and Sui. "Safe reinforcement learning in constrained Markov decision processes." ICML, 2020.
>
> - Thomas et al. "Safe reinforcement learning by imagining the near future." NeurIPS. 2021.
>
> A similar reasoning applies to Assumption 6, which ensures the asymptotic normality of the policy parameter estimator. Even when asymptotic normality does not strictly hold, the approximate consistency
> $|J(\pi_{\hat\theta,z}) - z| \le \varepsilon(z) + o_P(1)$
> can be established under weaker conditions (e.g., Theorem 5.14 in Van der Vaart [51]). This level of consistency is sufficient for characterizing the safe set used in Assumption 7.
>
>
> ### [Weakness 4] High-dimensional or multiple constraints setting
>
> We agree with the reviewer that Srinivas et al. (2014) specifically tackled a high-dimensional setting. However, safe exploration literature, such as Sui et al. (2015), Berkenkamp et al. (2016), and Turcehtta et al. (2016), has applied extensions of GP-UCB primarily in low-dimensional input spaces, at least empirically:
>
> - Sui et al. "Safe exploration for optimization with Gaussian processes." ICML. 2015.
>
> - Turchetta et al. "Safe exploration in finite Markov decision processes with Gaussian processes." NeurIPS. 2016.
>
> In our case, PLS is designed for sequential decision-making problems with a single safety constraint, which covers a broad class of safe RL problems encountered in practice.
> As the reviewer notes, however, extending to multiple constraints or high-dimensional return spaces presents important and non-trivial challenges. Our current PLS method may face computational limitations in such settings, and addressing them may require fundamentally different algorithmic designs. We will include additional discussion of these limitations and future directions in the camera-ready version.

---

> > ### Comment · Reviewer_XNH5 · 2025-08-07
> > **Response to the authors**
> >
> > I am satisfied with the author's response and maintain my slight positive rating. I would still suggest that the paper requires several layers of re-writing, especially around the assumptions and the theoretical contributions (explaining them more with greater clarity).

---

### Official Review · Reviewer_hNWo · 2025-07-03

**Clarity:** 2
**Significance:** 3
**Originality:** 3
**Rating:** 4
**Confidence:** 3

**Summary:**

This paper introduces ​Provably Lifetime Safe RL (PLS)​, a novel method addressing the critical challenge of ensuring policy safety throughout both learning and deployment phases in safe reinforcement learning. PLS integrates ​offline policy learning​ via CDT and safe online deployment​ using Gaussian Processes. Theoretically, the authors justify GP usage and establish safety guarantees during deployment.

**Questions:**

Regarding Section 6.2, could you please provide a more intuitive explanation and discussion of the design choices presented?

**Ethical Concerns:**

["NO or VERY MINOR ethics concerns only"]

**Final Justification:**

The authors clarification indeed addresses my concerns. The justification that PLS as a ​meta-algorithm​ and highlight its robustness beyond initial theoretical assumptions.

I think it's essential for the ​key arguments​ regarding the theoretical assumptions (especially the nature of Assumptions and the meta-algorithm flexibility) are incorporated into the paper's main text or relevant sections, ensuring readers benefit from these clarifications without needing the rebuttal.

**Limitations:**

Yes

**Quality:**

3

**Strengths And Weaknesses:**

Strengths:​​
To ensure safety while maximizing reward throughout the entire process from learning to execution, the policy is formulated as a return-conditioned policy, where the return comprises both reward and cost. This policy is trained offline via supervised learning. During inference, Gaussian Processes are effectively employed to model the relationship between the target return and the actual return. This model is then used to determine the optimal target return that guides the policy toward safe and high-reward behavior.

​This paper unifies offline safe RL and Bayesian optimization for end-to-end safety, addressing a fundamental gap in safe RL literature. The core innovation of this work lies in the integration of offline reinforcement learning with Gaussian Processes, offering a new design that bridges data-driven policy learning and uncertainty-aware return optimization.

The empirical results are compelling, demonstrating the effectiveness of the proposed method.

Weaknesses:​​

This paper relies a series of assumptions, such as near-deterministic transitions (Assumption 1) and initial safe target returns (Assumption 7), which may not hold in highly stochastic environments. The performance of proposed method hinges on offline dataset coverage and quality; distributional shifts may degrade GP predictions.

The theoretical sections use overly complex notation, which hampers readability. Additionally, some notations are reused for different concepts, for example, $\pi$ is used to denote both the policy and circle ratio. This reuse can lead to confusion and should be avoided. Simplifying and standardizing the notation would greatly enhance the clarity of the presentation.

---

> ### Author Rebuttal · Authors · 2025-07-29
>
> We deeply appreciate the reviewer hNWo for helpful and thoughtful comments and questions. We will first address the comments in Weaknesses and then answer the Question.
>
> > **[Weakness 1]** This paper relies a series of assumptions, such as near-deterministic transitions (Assumption 1) and initial safe target returns (Assumption 7), which may not hold in highly stochastic environments. The performance of proposed method hinges on offline dataset coverage and quality; distributional shifts may degrade GP predictions.
>
> Thank you for raising this important point. We agree that assumptions such as near-deterministic transitions (Assumption 1) and the existence of an initial safe target return (Assumption 7) may not hold in highly stochastic or low-coverage settings. However, Assumptions 1-6 are introduced to conduct our theoretical analysis particularly for modeling Constrained Decision Transformers (CDTs), but they are not inherent to the general structure of PLS.
>
> In particular, while Assumption 1 (near-deterministic transitions) is strong, it is more general than the one used in Brandfonbrener et al. [9], where exact deterministic transitions (i.e., $\delta = 0$) are assumed. Our formulation allows for a $\delta$-neighborhood around deterministic dynamics, which permits small stochastic perturbations and thus covers a broader class of environments. Moreover, similar (near-)deterministic assumptions are common in the safe RL literature, as seen in:
>
> - Berkenkamp et al. "Safe model-based reinforcement learning with stability guarantees." NeurIPS, 2017.
>
> - Wachi and Sui. "Safe reinforcement learning in constrained Markov decision processes." ICML, 2020.
>
> - Thomas et al. "Safe reinforcement learning by imagining the near future." NeurIPS. 2021.
>
> Our PLS is designed as a meta-algorithm and can be empirically paired with any return-conditioned offline RL where performance can be approximated by a Gaussian Process (GP) as a function of target returns. If future policy architectures relax or eliminate assumptions like near-determinism, PLS can incorporate them directly and may benefit from improved generalization and theoretical guarantees.
>
> Regarding offline dataset coverage and distribution mismatch, we acknowledge that these factors can affect GP prediction accuracy. However, a key feature of PLS is that it actively learns the discrepancy between target and actual returns during online deployment, using uncertainty-aware safe exploration and reward maximization. This mitigates the risk of unsafe behavior due to model miscalibration or distributional shifts.
>
> > **[Weakness 2]** The theoretical sections use overly complex notation, which hampers readability. Additionally, some notations are reused for different concepts, for example, $\pi$ is used to denote both the policy and circle ratio. This reuse can lead to confusion and should be avoided. Simplifying and standardizing the notation would greatly enhance the clarity of the presentation.
>
> Thank you for highlighting the notational ambiguity. We acknowledge the reuse of the symbol $\pi$ and have included a disclaimer regarding this on line 257 of the original submission. Nonetheless, we agree that the reuse may still be confusing. In the camera-ready version, we will revise the notation to avoid this overlap and improve clarity.
>
> We believe precise notation is essential for conveying the theoretical content accurately, and we have already attempted to manage the overall complexity by moving detailed mathematical notations to the appendix. If there are specific notations that felt unclear, we would be grateful for your suggestions and happy to incorporate them. In the camera-ready version, we will include a nomenclature table in the appendix. For example,
>
> | Symbol | Meaning | Section |
> |--------|---------|------|
> | $\pi$   | context-dependent policy | 3 |
> | $\pi_\text{circle}$   | circle ratio ($\approx$ 3.14) |6 |
> | $\beta_R$ | return-conditioned behavior policy | 4 |
> | $b$     | safety budget | 3 |
> | $\vdots$ | $\vdots$ | $\vdots$ |
> | $R, G$    | target reward / cost | 3 |
>
> > **[Question]** Regarding Section 6.2, could you please provide a more intuitive explanation and discussion of the design choices presented?
>
> Our goal in Section 6.2 is to find the best target returns $z = (R,G)$ that maximize reward while satisfying the safety constraint. We achieve this through a two-phase process guided by GP uncertainty estimates:
>
> - Pessimism for safety: To ensure that selected targets are safe, we evaluate the safety cost using the GP’s upper confidence bound. This pessimistic estimate accounts for worst-case scenarios. A new target return is considered only if this worst-case cost remains below the safety threshold $b$.
>
> - Optimism for reward: To encourage discovering high-performing policies, we use the GP’s upper confidence bound for the reward. This optimistic estimate helps explore targets with the highest potential reward return.
>
> In the safe exploration phase, we first explore the most uncertain but still verifiably safe target returns to efficiently expand the safe region. Also, in the reward maximization phase, we switch to selecting targets that maximize the optimistic reward estimate within the safe region.

---

> > ### Comment · Reviewer_hNWo · 2025-08-07
> > **Thanks for the rebuttal**
> >
> > Thanks for the rebuttal. The authors clarification indeed addresses my concerns. The justification that PLS as a ​meta-algorithm​ and highlight its robustness beyond initial theoretical assumptions.
> >
> > I think it's essential for the ​key arguments​ regarding the theoretical assumptions (especially the nature of Assumptions and the meta-algorithm flexibility) are incorporated into the paper's main text or relevant sections, ensuring readers benefit from these clarifications without needing the rebuttal. Based on this rebuttal, I am inclined to ​raise my score.

---

### Note · Authors · 2025-08-13

Dear Area Chair and Reviewers,

We appreciate the constructive feedback and engagement from all reviewers and the Area Chair. The discussion helped clarify our contributions and address concerns. Three reviewers (hNWo, XNH5, A6AP) expressed satisfaction after the rebuttal, with Reviewer hNWo increasing their score. Reviewer njCE acknowledged the value of the combined theoretical and empirical approach and stated that many of their questions and sources of confusion were resolved, though still noted limited empirical scope.

Our core contributions are:

- The first provably end-to-end safe RL framework, guaranteeing safety from offline training through online deployment.

- A strong theoretical foundation (Theorems 1-3) that formally links target and actual returns, proving high-probability safety and near-optimality.

- A consistent empirical advantage, with PLS being the only method to satisfy safety constraints across all 9 tested tasks while achieving competitive rewards.

- High sample efficiency and safety in deployment, significantly outperforming baselines as shown in our discussion with Reviewer A6AP.

During the discussion phase, we addressed specific points raised by the reviewers:

- Assumptions: Several are for theoretical tractability rather than being inherent to PLS. The method performs well when these are relaxed, and the meta-algorithm design allows other policy architectures to be substituted.

- Clarity: We will revise notation, add a nomenclature table, and rewrite sections of the theory for readability.

- Empirical scope: We focused on the CCAC benchmark set for a direct comparison with the most relevant SOTA baseline. Extending to other domains is a natural next step.

- Extensions: While our current work keeps the policy network fixed online, PLS can serve as a basis for safe offline-to-online learning with theoretical guarantees.

We believe PLS addresses a long-standing open problem in safe RL: achieving safety guarantees throughout learning and operation without significantly sacrificing reward. We have incorporated the reviewers’ suggestions into the final version to improve clarity and transparency.

Best regards,

Authors

---

### Decision · Program_Chairs · 2025-09-17

**Decision:**

Accept (poster)

**Comment:**

The paper proposes a method called Provably Lifetime Safe RL (PLS) to ensure a reinforcement learning policy is safe throughout its entire lifecycle, from training to deployment.

### Key Strengths:
- The combination of offline RL with online safe optimization using Gaussian Processes was seen as novel and well-motivated.
- The paper's main strength was its rigorous theoretical analysis, providing provable safety guarantees, which addresses a significant gap in safe RL.
- Empirically, the PLS method consistently outperformed baselines by successfully maintaining safety across all experiments while achieving competitive rewards.

### Weakness:
- Several reviewers noted that the theoretical guarantees rely on strong assumptions (e.g., near-deterministic environments) that may not hold in real-world scenarios.
- The most significant point of contention was the limited number of experiments. One reviewer (njCE) rated the paper "borderline reject" primarily because the evaluation wasn't extensive enough, even though the theory was solid.
- Some reviewers found the theoretical sections and notation complex and hard to read.
- The method only optimizes the target returns online, not the policy network itself, which limits its ability to adapt from new online data.

### Author Rebuttal:
The authors provided a comprehensive rebuttal that successfully addressed most concerns.

### Conclusion:
The final consensus among the reviewers and the AC was positive